# An in vivo reporter for tracking lipid droplet dynamics in transparent zebrafish

Dianne Lumaquin[1,2†], Eleanor Johns[1,3†], Emily Montal[1], Joshua M Weiss[1,2], David Ola[1], Abderhman Abuhashem[2,4], Richard M White[1]*

[1]Cancer Biology and Genetics Program, Memorial Sloan Kettering Cancer Center, New York, United States; [2]Weill Cornell/Rockefeller/Sloan Kettering Tri-Institutional MD-PhD Program, New York, United States; [3]Gerstner Sloan Kettering Graduate School of Biomedical Sciences, Memorial Sloan Kettering Cancer Center, New York, United States; [4]Developmental Biology Program, Memorial Sloan Kettering Cancer Center, New York, United States

**Abstract** Lipid droplets are lipid storage organelles found in nearly all cell types from adipocytes to cancer cells. Although increasingly implicated in disease, current methods to study lipid droplets in vertebrate models rely on static imaging or the use of fluorescent dyes, limiting investigation of their rapid in vivo dynamics. To address this, we created a lipid droplet transgenic reporter in whole animals and cell culture by fusing tdTOMATO to Perilipin-2 (PLIN2), a lipid droplet structural protein. Expression of this transgene in transparent *casper* zebrafish enabled in vivo imaging of adipose depots responsive to nutrient deprivation and high-fat diet. Simultaneously, we performed a large-scale in vitro chemical screen of 1280 compounds and identified several novel regulators of lipolysis in adipocytes. Using our *Tg(-3.5ubb:plin2-tdTomato)* zebrafish line, we validated several of these novel regulators and revealed an unexpected role for nitric oxide in modulating adipocyte lipid droplets. Similarly, we expressed the PLIN2-tdTOMATO transgene in melanoma cells and found that the nitric oxide pathway also regulated lipid droplets in cancer. This model offers a tractable imaging platform to study lipid droplets across cell types and disease contexts using chemical, dietary, or genetic perturbations.

*For correspondence:
whiter@mskcc.org

†These authors contributed equally to this work

## Introduction

Lipid droplets are cellular organelles that act as storage sites for neutral lipids and are key regulators of cellular metabolism (*Wilfling et al., 2013*). Lipid droplets are present in most cell types and are characterized by a monophospholipid membrane surrounding a hydrophobic lipid core (*Tauchi-Sato et al., 2002*; *Wilfling et al., 2013*). Cells maintain energetic homeostasis and membrane formation through the regulated incorporation and release of fatty acids and lipid species from the lipid droplet core (*Kurat et al., 2009*; *Kuerschner et al., 2008*; *Fujimoto et al., 2007*; *Zimmermann et al., 2004*). Importantly, lipid droplets can assume various functions during cellular stress through the sequestration of potentially toxic lipids and misfolded proteins (*Bailey et al., 2015*; *Listenberger et al., 2003*; *Vevea et al., 2015*; *Fei et al., 2009*), maintenance of energy and redox homeostasis (*Liu et al., 2015*; *Liu et al., 2017*), regulation of fatty acid transfer to the mitochondria for β-oxidation (*Rambold et al., 2015*; *Herms et al., 2015*), and the maintenance of endoplasmic reticulum (ER) membrane homeostasis (*Chitraju et al., 2017*; *Bosma et al., 2014*; *Velázquez et al., 2016*; *Vevea et al., 2015*). Moreover, a recent study demonstrated that lipid droplets actively participate in the innate immune response (*Bosch et al., 2020*) and, conversely, can be hijacked by infectious agents like hepatitis C virus to facilitate viral replication (*Barba et al., 1997*; *Miyanari et al., 2007*; *Vieyres et al., 2020*). The role of lipid droplets in metabolic homeostasis and cellular stress is critical across multiple cell types and has also been increasingly implicated in

**eLife digest** Organisms need fat molecules as a source of energy and as building blocks, but these 'lipids' can also damage cells if they are present in large amounts. Cells guard against such toxicity by safely sequestering lipids in specialized droplets that participate in a range of biological processes. For instance, these structures can quickly change size to store or release lipids depending on the energy demands of a cell.

It is possible to image lipid droplets – using, for example, dyes that preferentially stain fat – but often these methods can only yield a snapshot: tracking lipid droplet dynamics over time remains difficult. Lumaquin, Johns et al. therefore set out to develop a new method that could label lipid droplets and monitor their behaviour 'live' in the cells of small, transparent zebrafish larvae.

First, the fish were genetically manipulated so that a key protein found in lipid droplets would carry a fluorescent tag: this made the structures strongly fluorescent and easy to track over time. And indeed, Lumaquin, Johns et al. could monitor changes in the droplets depending on the fish diet, with the structures getting bigger when the animal received rich food, and shrinking when resources were scarce. Finally, experiments were conducted to screen for compounds that could lead to lipids being released in fat cells. The new imaging technique was then used to confirm the effect of these molecules in live cells, revealing an unexpected role for a signalling molecule known as nitric oxide, which also turned out to be regulating lipid droplets in cancerous cells. Further work then showed that drugs affecting nitric oxide could modulate lipid droplet size in both normal and tumor cells.

This work has validated a new method to study the real-time behavior of lipid droplets and their responses to different stimuli in living cells. In the future, Lumaquin, Johns et al. hope that the technique will help to shed new light on how lipids are involved in both healthy and abnormal biological processes.

cancer. For example, lipid droplets can act as a storage pool in cancer cells after they take up lipids from extracellular sources, including adipocytes (*Kuniyoshi et al., 2019*; *Nieman et al., 2011*; *Zhang et al., 2018*).

Lipid droplets are ubiquitous across most cell types; however, they are essential to the function of adipocytes in regulating organismal energy homeostasis (*Zimmermann et al., 2004*; *Bergman et al., 2001*). White adipocytes, which contain a large unilocular lipid droplet, can be readily labeled by lipophilic dyes (*Minchin and Rawls, 2017a*; *Fam et al., 2018*). However, in vivo imaging of lipid droplets, in adipocytes or other cell types, is currently highly limited. Understanding the dynamics of lipid droplets in vivo, rather than in fixed tissues, is important since the size of the lipid droplet can change very rapidly in response to fluctuating metabolic needs (*Bosch et al., 2020*; *Fam et al., 2018*). In mice, much of adipose tissue imaging utilizes tissue fixation and sectioning, which can fail to preserve key aspects of the tissue structure (*Berry et al., 2014*; *Xue et al., 2010*). Whole-mount imaging approaches in mice can be combined with adipocyte-specific promoters; however, these methods still require tissue dissection and can be limited by tissue thickness (*Berry and Rodeheffer, 2013*; *Chi et al., 2018*).

Zebrafish offer a tractable model to address these limitations given the ease of high-throughput imaging of live animals. This is especially true with the availability of relatively transparent strains such as *casper*, which allows for detailed in vivo imaging without the need for fixation of the animal (*White et al., 2008*). Although less well studied than other vertebrates, zebrafish adipose tissue is highly similar to mammalian white adipose tissue, and a detailed work has classified the timing, dynamics, and location of zebrafish adipose tissue development (*Minchin and Rawls, 2017a*). Current imaging approaches using lipophilic fluorescent dyes or analogs in vivo have advanced our understanding of lipid droplets in adipocytes and other cell types (*Minchin and Rawls, 2017a*; *Minchin et al., 2018*; *Minchin and Rawls, 2017b*; *Carten et al., 2011*; *Farber et al., 2001*; *Otis and Farber, 2016*); however, these methods can require extensive and repeated staining, which may restrict the ability to read out dynamic changes over time. Furthermore, fluorescent dyes such as BODIPY and NileRed have limitations in their specificity for the lipid droplet (*Daemen et al., 2016*). Finally, although a probe-free imaging approach to study subcellular lipid populations has

been recently described (*Høgset et al., 2020*), this method is restricted to early-stage zebrafish, which would fail to capture post-embryonic cell populations and tissues, including adipocytes.

Here, we report the development of an in vivo lipid droplet reporter using a *-3.5ubb:plin2-tdTomato* transgene in the *casper* strain. To date, transgenic lipid droplet reporters have been restricted to cell culture systems and invertebrate model organisms such as *Caenorhabditis elegans* (*C. elegans*) and *Drosophila* (*Targett-Adams et al., 2003*; *Beller et al., 2010*; *Liu et al., 2014*; *Liu et al., 2015*) although a similar approach in zebrafish was recently described while this manuscript was in review (*Wilson et al., 2021*). We demonstrate that the *-3.5ubb:plin2-tdTomato* reporter faithfully marks the lipid droplet, which enables robust in vivo imaging. We show that this reporter can be applied to visualize adipocytes and to monitor adipose tissue remodeling in response to dietary and pharmacologic perturbations. Furthermore, we report the discovery of novel pharmacologic regulators of adipocyte lipolysis such as nitric oxide and demonstrate that several of these compounds can modulate adipose tissue area in our in vivo system. To facilitate the study of lipid droplets in novel contexts outside of adipocytes, we also generated a zebrafish melanoma cell line (ZMEL) (*Heilmann et al., 2015*) expressing *-3.5ubb:plin2-tdTomato* (ZMEL-LD). We confirm that this cell line can be used to monitor changes in lipid droplet production in response to both known and novel regulators of lipolysis. We anticipate that these models will be highly valuable as a high-throughput imaging platform to investigate lipid droplets in both adipose tissue biology and disease contexts such as cancer.

## Results

### An in vivo lipid droplet reporter using a PLIN2-tdTOMATO fusion transgene

To create a specific fluorescent reporter for lipid droplets in zebrafish, we fused *tdTomato* to the 3' end of the *plin2* cDNA. We chose *plin2* because it is a well-known lipid droplet-associated protein that is ubiquitously expressed on lipid droplets across cell types (*Olzmann and Carvalho, 2019*). We generated stable transgenic zebrafish expressing *-3.5ubb:plin2-tdTomato* and sought to validate whether the construct faithfully marks lipid droplets (*Figure 1A*). White adipocytes are fat cells known for their large unilocular lipid droplet (*Fujimoto and Parton, 2011*; *Heid et al., 2014*), so we expected expression of the PLIN2-tdTOMATO fusion protein on the surface of the adipocyte lipid droplet (*Figure 1A*). Since the adipocyte lipid droplet occupies the majority of space in the cell (*Fujimoto et al., 2020*), existing methods to visualize zebrafish adipocytes rely on lipophilic dyes and lipid analogs, which incorporate into the lipid droplet (*Zhang et al., 2018*). Thus, in addition to labeling individual lipid droplets, we reasoned that the PLIN2-tdTOMATO fusion protein can also function as a reporter for adipocytes since these cells would have the largest and unilocular lipid droplets.

In adult zebrafish, subcutaneous adipocytes are known to reside proximally to the tail fin (*Minchin and Rawls, 2017a*). When we imaged 6-month-old adult *Tg(-3.5ubb:plin2-tdTomato)* zebrafish, we detected PLIN2-tdTOMATO expression in the zebrafish tail fin adipocytes, which colocalized with BODIPY staining (*Figure 1B,C*). Lipophilic dyes such as BODIPY stain the lipid-rich core of the lipid droplet while lipid droplet resident proteins, such as PLIN2, localize to the lipid droplet membrane (*Zhang et al., 2018*). As expected, higher magnification images of tail adipocytes revealed that PLIN2-tdTOMATO expression was on the outside of the lipid droplet, whereas the BODIPY staining was on the interior of each droplet in the adipocyte (*Figure 1D*). Similarly, immunohistochemistry (IHC) on the *Tg(-3.5ubb:plin2-tdTomato)* zebrafish tail fin showed that adipocytes express tdTOMATO (*Figure 1E*). Taken together, this data demonstrates that the PLIN2-tdTOMATO fusion protein functions as a fluorescent lipid droplet reporter that can be applied to visualize adipocytes in vivo.

### The *Tg(-3.5ubb:plin2-tdTomato)* is an in vivo reporter for visceral adipocytes

Visceral adipose tissue, otherwise known as abdominal fat, plays an important role in metabolism and participates in pathological processes of obesity, aging, and metabolic syndromes (*Tchernof and Després, 2013*). Because PLIN2-tdTOMATO labeled subcutaneous adipocytes in the

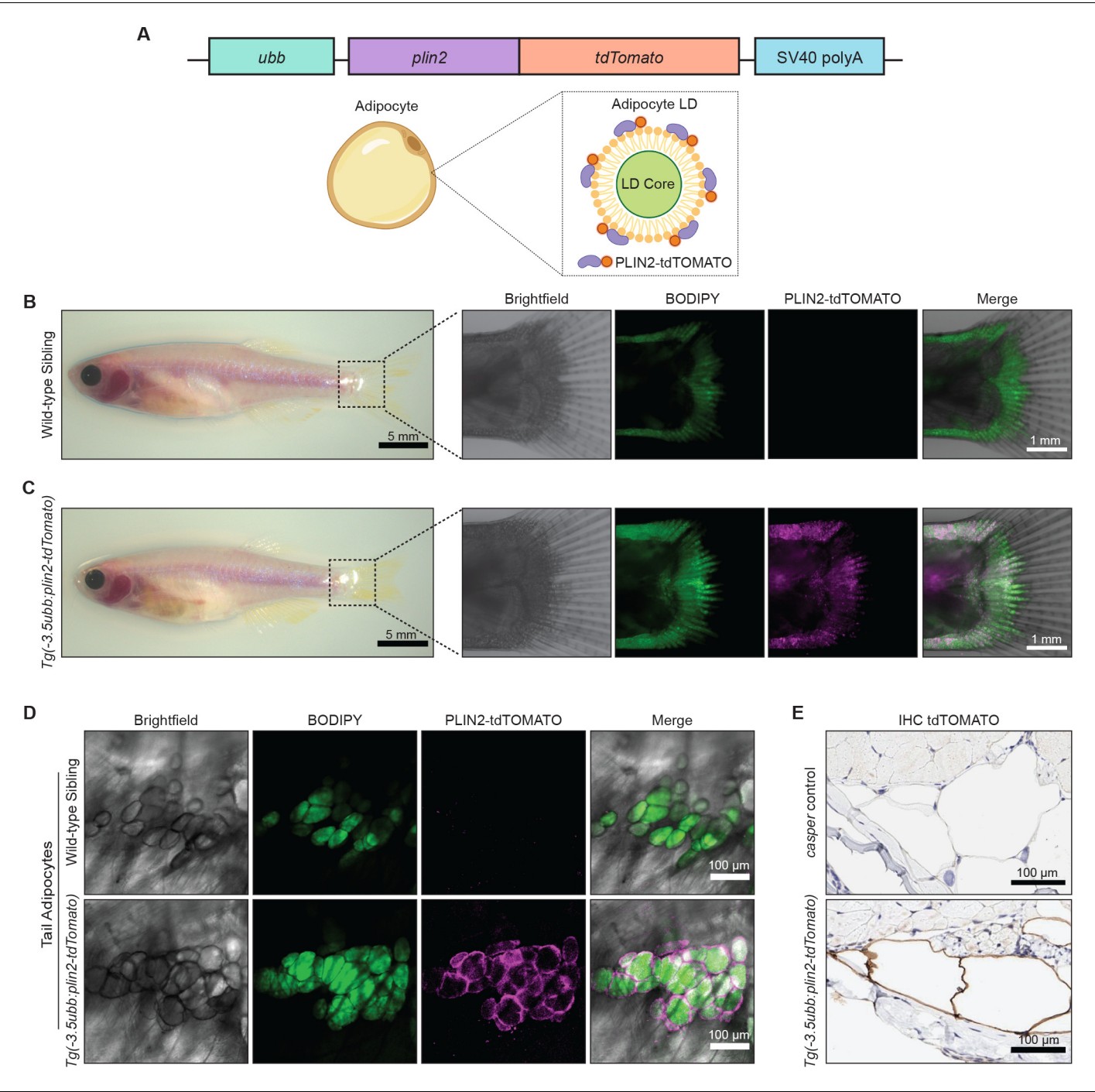

**Figure 1.** An in vivo lipid droplet reporter using a PLIN2-tdTOMATO fusion transgene. (**A**) Schematic of *ubb:plin2-tdTomato* construct injected into zebrafish and an adipocyte lipid droplet labeled with PLIN2-tdTOMATO fusion protein. Widefield microscope images of adult (**B**) wild-type sibling and (**C**) *Tg(-3.5ubb:plin2-tdTomato)* zebrafish. Box shows zoomed images of the fish tail with panels for brightfield, BODIPY, PLIN2-tdTOMATO, and merge. (**D**) Confocal images of fish tail adipocytes of adult wild-type sibling and *Tg(-3.5ubb:plin2-tdTomato)* zebrafish. Panels show brightfield, BODIPY, PLIN2-tdTOMATO, and merge. (**E**) Adult *casper* and *Tg(-3.5ubb:plin2-tdTomato)* zebrafish tails were fixed and immunohistochemistry was conducted for tdTOMATO expression of tail adipocytes.

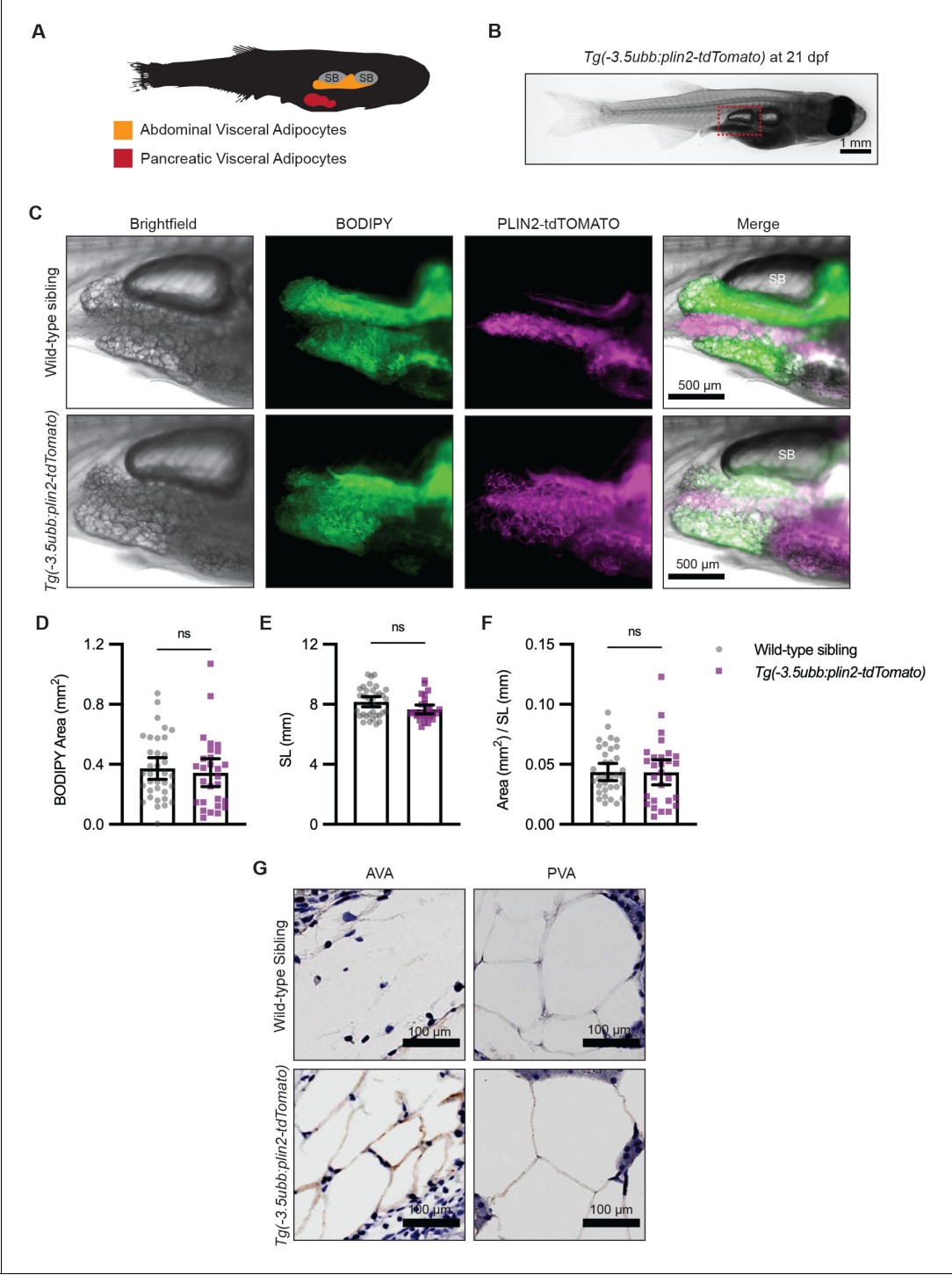

**Figure 2.** *Tg(-3.5ubb:plin2-tdTomato)* is an in vivo reporter for visceral adipocytes. (**A**) Schematic of visceral adipose tissue development in the 21 days post-fertilization (dpf) zebrafish. Abdominal visceral adipocytes (orange) develop around the swim bladder (gray) and pancreatic visceral adipocytes (red) develop ventrally around the pancreas. (**B**) Brightfield image of *Tg(-3.5ubb:plin2-tdTomato)* at 21 dpf. Red box indicates position of higher magnification images to visualize abdominal visceral adipocytes. (**C**) Widefield microscope images of 21 dpf wild-type *casper* and *Tg(-3.5ubb: plin2-tdTomato)* visceral adipocytes around the posterior swim bladder (SB) costained with BODIPY. Panels show brightfield, BODIPY, tdTOMATO, and merge. BODIPY stained adipose tissue was imaged and analyzed for (**D**) area, (**E**) standard length, and (**F**) area/standard length. Points indicate individual fish for N= 3 independent experiments; wild-type sibling, Data values for Figure 2D-F. n=35; *Tg(-3.5ubb:plin2-tdTomato)*, n= 28. Bars indicate mean and SEM. Significance calculated via Mann-Whitney test. (**G**) 21 dpf wild-type *casper* and *Tg(-3.5ubb:plin2-tdTomato)* zebrafish were fixed and immunohistochemistry conducted for tdTOMATO expression of abdominal (AVA) and pancreatic visceral adipocytes (PVA).

*Figure 2 continued on next page*

*Figure 2 continued*

The online version of this article includes the following source data and figure supplement(s) for figure 2:

**Source data 1.** Data values for Figure 2D-F.

**Figure supplement 1.** Adipose tissue development in wild-type and *Tg(-3.5ubb-plin2-tdTomato)* zebrafish.

**Figure supplement 1—source data 1.** Data values for Figure 2-figure supplement 1A.

adult zebrafish tail fin, we wondered whether we could use the *Tg(-3.5ubb:plin2-tdTomato)* zebrafish to visualize other adipose depots in vivo such as visceral adipocytes. In 21 days post-fertilization (dpf) zebrafish, visceral adipose tissue is composed of abdominal and pancreatic visceral adipocytes predominantly located on the right flank near the swim bladder (*Figure 2A*; *Minchin and Rawls, 2017a*). To determine whether *Tg(-3.5ubb:plin2-tdTomato)* visceral adipocytes express PLIN2-tdTO-MATO, we imaged around the swim bladder of zebrafish where we expected development of abdominal visceral adipocytes (*Figure 2B*). Visceral adipocytes visualized in brightfield demonstrate colocalization of PLIN2-tdTOMATO and BODIPY, as we observed for subcutaneous adipocytes (*Figure 2C*). IHC confirmed that the abdominal and visceral adipocytes of *Tg(-3.5ubb:plin2-Tomato)* express tdTOMATO (*Figure 2G*).

Previous studies have shown that overexpression of perilipin proteins alters lipid accumulation in adipocytes (*Sawada et al., 2010*). To test whether constitutive expression of our PLIN2-tdTOMATO transgenes altered adiposity, we compared the visceral adipose tissue of *Tg(-3.5ubb:plin2-tdTo-mato)* fish to their wild-type siblings. Using BODIPY staining, we did not detect differences in visceral adipose tissue area (*Figure 2D*). In addition to dpf, standard length can indicate developmental progress in zebrafish (*Parichy et al., 2009*). Similarly, we found no difference in standard lengths between wild-type siblings and *Tg(-3.5ubb:plin2-tdTomato)* at 21 dpf (*Figure 2E*). We normalized visceral adipose tissue area to standard length, similar to a Body Mass Index (BMI) in mammals, and did not detect differences between wild-type and *Tg(-3.5ubb:plin2-tdTomato)* fish (*Figure 2F*). To comprehensively assess whether constitutive PLIN2-tdTOMATO expression alters adiposity, we imaged adipose depots of wild-type and *Tg(-3.5ubb:plin2-tdTomato)* zebrafish from 5 dpf larvae to adult fish. Using BODIPY staining to visualize adipose tissue, we observed similar development of the major adipose depots between wild-type and *Tg(-3.5:ubbplin2-tdTomato)* zebrafish (*Figure 2—figure supplement 1A–B*). Furthermore, we detected PLIN2-tdTOMATO expression at the corresponding time points and adipose depots within *Tg(-3.5ubb:plin2-tdTomato)* (*Figure 2—figure supplement 1A–B*). Thus, the *Tg(-3.5ubb:plin2-tdTomato)* zebrafish faithfully reca-pitulates normal adipose tissue development.

## Diet and pharmacologically induced reduction in visceral adipose tissue area

After confirming that we could image visceral adipose tissue in *Tg(-3.5ubb:plin2-tdTomato)*, we wanted to test whether this could be a tractable platform to image adipose tissue remodeling. We first verified whether we could use *Tg(-3.5ubb:plin2-tdTomato)* to track reduction in visceral adipos-ity. Fasting is a well-known mechanism for reducing adiposity, since it will induce lipolysis and lead to a reduction in the size of the adipocyte lipid droplet (*Henne et al., 2018*; *Longo and Mattson, 2014*; *Rambold et al., 2015*; *Tang et al., 2017*). To test this, wild-type and *Tg(-3.5ubb:plin2-tdTo-mato)* zebrafish were fed or fasted for 7 days and then imaged to measure standard length and adi-pose tissue area (*Figure 3A*). As expected, we observed a reduction in the BODIPY-stained visceral adipose tissue in both wild-type and *Tg(-3.5ubb:plin2-tdTomato)* fasted zebrafish (*Figure 3B*). Simi-larly, we measured a significant reduction in adipose tissue area, standard length, and normalized area to standard length between fed and fasted fish (*Figure 3C,D,E*). Furthermore, we did not see differences in adiposity or standard length between wild-type and *Tg(-3.5ubb:plin2-tdTomato)* fish within the same diet (*Figure 3C,D,E*). This suggests that the transgenic expression of PLIN2-tdTO-MATO reflects adipose tissue remodeling consistent with wild-type fish.

Combined with the capacity for high-throughput in vivo imaging in zebrafish, we sought to use *Tg(-3.5ubb:plin2-tdTomato)* as a model to study lipid droplet dynamics in visceral adipocytes. One challenge we encountered was the autofluorescence from the zebrafish intestinal loops and gallblad-der present in the tdTOMATO and GFP channels (*Figure 3F*). To remove background fluorescence,

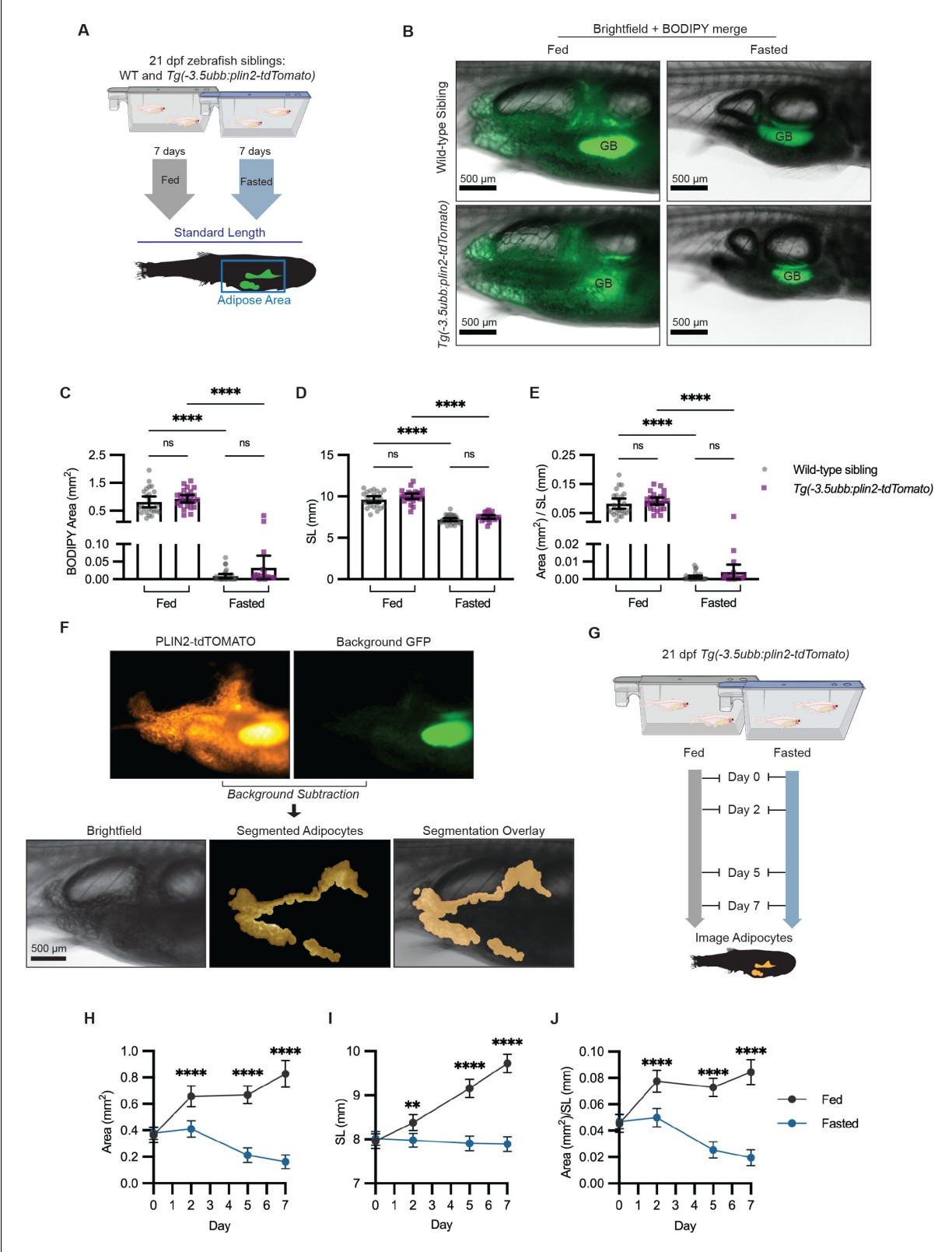

**Figure 3.** Fasting reduces visceral adipose tissue area. (**A**) Schematic of experimental set-up for fasting experiment. 21 days post-fertilization (dpf) wild-type *casper* and *Tg(-3.5ubb:plin2-tdTomato)* zebrafish were fed or fasted for 7 days and imaged to measure standard length and adipose area. (**B**) Representative images of zebrafish fed or fasted after 7 days. Panels show merged images of brightfield and BODIPY-stained visceral adipose tissue. BODIPY-stained adipocytes were imaged to measure (**C**) area, (**D**) standard length, and (**E**) area/standard length. Data points indicate individual fish for

*Figure 3 continued on next page*

*Figure 3 continued*

N = 3 independent experiments; bars indicate mean and 95% CI. Fed wild-type n = 24; fed *Tg(-3.5ubb:plin2-tdTomato)* n = 24; fasted wild-type n = 29; fasted *Tg(-3.5ubb:plin2-tdTomato)* n = 20. Significance calculated via Kruskal-Wallis with Dunn's multiple comparisons test; ****p<0.0001. (F) Representative image of computational segmentation of *Tg(-3.5ubb:plin2-tdTomato)* adipocytes. PLIN2-tdTOMATO was background subtracted with GFP fluorescence. Bottom panels show brightfield, segmented adipocytes, and segmentation overlaid on brightfield. (G) Schematic of experimental set-up for repeated imaging of 21 dpf *Tg(-3.5ubb:plin2-tdTomato)* zebrafish, which were fed or fasted for 7 days. Adipose tissue was imaged and analyzed for (H) area, (I) standard length, and (J) area/standard length. Points indicate mean and error bars indicate 95% CI for N = 3 independent experiments; by day 7, fed n = 46 and fasted n = 57. Significance calculated via Mann-Whitney test; **p<0.01, ****p<0.0001.

The online version of this article includes the following source data and figure supplement(s) for figure 3:

**Source data 1.** Data values for Figure 3C-E.
**Source data 2.** Data values for Figure 3H-J.
**Figure supplement 1.** Forskolin reduces visceral adipose tissue area.
**Figure supplement 1—source data 1.** Data values for Figure 3-figure supplement 1B-D.

we developed an image analysis pipeline in MATLAB to segment the visceral adipocytes in juvenile *Tg(-3.5ubb:plin2-Tomato)* zebrafish (*Figure 3F*). Next, we combined our image analysis pipeline with the ability to do repeated imaging in zebrafish to more granularly quantify adipose tissue remodeling in fed or fasted *Tg(-3.5ubb:plin2-tdTomato)* fish (*Figure 3G*). We saw the expected increase in adipose tissue area, standard length, and normalized area to standard length over 7 days in fed *Tg(-3.5ubb:plin2-tdTomato)* fish (*Figure 3H–J*). The standard lengths for fasted fish were significantly lower than those for fed fish and remained stable over 7 days, which we attribute to food restriction,

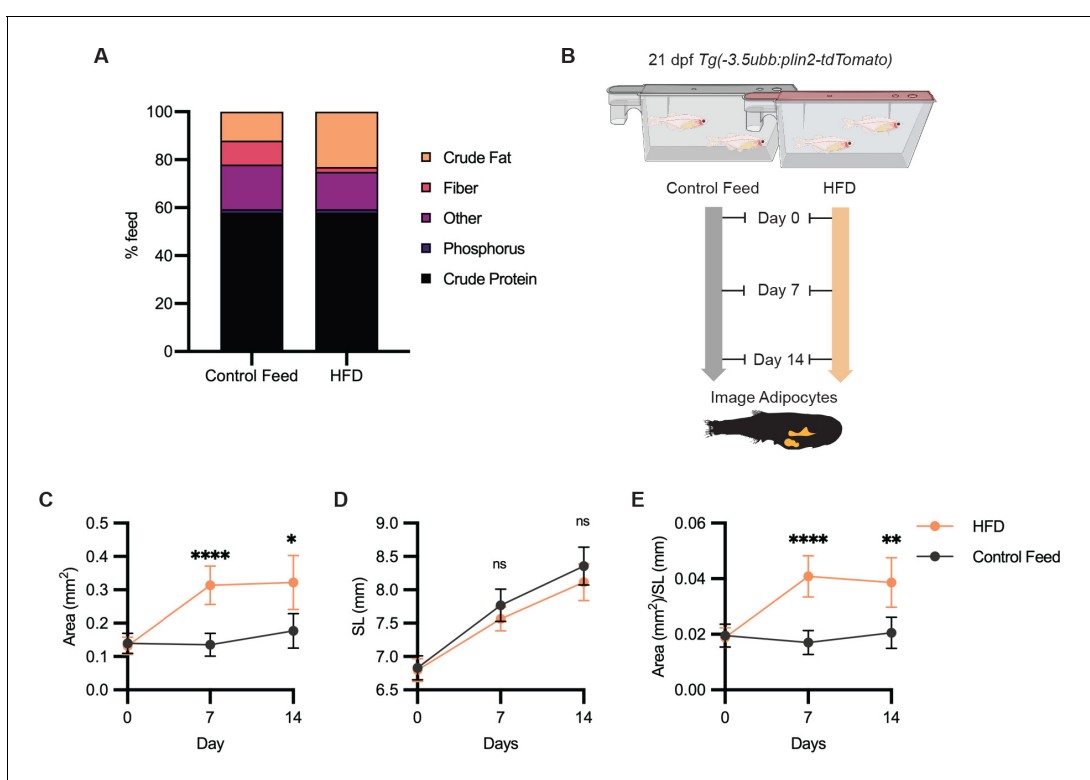

**Figure 4.** High-fat diet leads to specific enlargement of visceral adipose tissue. (A) Percent breakdown of nutritional content for control feed and high-fat diet (HFD). (B) Schematic of experimental set-up for HFD experiment. 21 days post-fertiization (dpf) *Tg(-3.5ubb:plin2-tdTomato)* zebrafish were fed control feed or HFD for 14 days and imaged to measure standard length and adipose area. Image analysis pipeline resulted in measurements of adipose tissue (C) area, (D) standard length, and (E) area/standard length. Points indicate mean and error bars indicate 95% CI for N = 3 independent experiments; by day 14, control feed n = 57; HFD n = 61. Significance calculated via Mann-Whitney test; *p<0.05, **p<0.01, ****p<0.0001.

The online version of this article includes the following source data for figure 4:

**Source data 1.** Data values for Figure 4 A-E.

disrupting zebrafish development (*Figure 3I*). Notably, we found that visceral adiposity was not reduced until after 2 days of fasting (*Figure 3H,J*).

In addition to fasting as a dietary perturbation, we also pharmacologically reduced adipose tissue. To achieve this, we used Forskolin, a drug that is known to induce lipolysis through cAMP signaling (*Litosch et al., 1982*). We treated juvenile zebrafish for 24 hr with either dimethyl sulfoxide (DMSO) or 5 µM Forskolin and imaged the adipocytes (*Figure 3—figure supplement 1A*). We detected a reduction in both the adipose tissue area and normalized area to standard length in the Forskolin-treated fish, but no differences in standard length (*Figure 3—figure supplement 1B–D*). Thus, *Tg(-3.5ubb:plin2-tdTomato)* can be used as an in vivo model to visualize adipocytes, with the benefits of avoiding staining steps and allowing for repeated imaging with high-throughput image analysis in zebrafish.

## High-fat diet leads to specific enlargement of visceral adipose tissue

Having shown that we could use *Tg(-3.5ubb:plin2-tdTomato)* to image and measure reduction in adipose tissue, we tested whether we can use our model to detect an increase in adiposity. Zebrafish have been used as a model for diet-induced obesity and share pathophysiological perturbations seen in mammals, but few studies have focused on architectural changes of visceral adipose tissue (*Chu et al., 2012*; *Landgraf et al., 2017*; *Oka et al., 2010*). We sought to determine if we could detect increases in visceral adiposity from a high-fat diet (HFD). We fed juvenile zebrafish with either control feed (12% crude fat) or HFD (23% crude fat) for 7 days and subsequently imaged the adipose tissue (*Figure 4A,B*). After a week of HFD feeding, we observed that HFD-fed fish developed significantly increased visceral adiposity compared to the fish fed with control feed (*Figure 4C,E*). Although less dramatic between 7 and 14 days of HFD, we found that significantly greater visceral adiposity persisted after 2 weeks of HFD (*Figure 4C,E*). Interestingly, we did not detect differences in the standard lengths of the control and HFD-fed fish, suggesting that this formulation of HFD leads to specific enlargement of visceral adipose tissue (*Figure 4E*). Our results demonstrate that *Tg(-3.5ubb:plin2-tdTomato)* is an effective and unique tool to visualize visceral adipose tissue remodeling induced by HFD, which can be widely applied to study obesity.

## A screen to discover novel compounds that modulate lipolysis and lipid droplets in vivo

To meet fluctuating nutritional needs of the cell, lipid droplets are remodeled through lipolysis to regulate lipid mobilization and metabolic homeostasis (*Krahmer et al., 2013*; *Olzmann and Carvalho, 2019*; *Paar et al., 2012*). As a major lipid depot for the body, white adipose tissue is critical to lipid availability and cycles through lipolytic flux in response to energy demands (*Duncan et al., 2007*). In disease contexts such as cancer, adipocytes undergoing lipolysis act as a lipid source for neighboring cancer cells (*Lengyel et al., 2018*). Adipocyte-derived lipids have been directly shown to promote cancer progression in ovarian (*Nieman et al., 2011*), breast (*Balaban et al., 2017*), and melanoma cancer cells (*Zhang et al., 2018*). Due to growing evidence of adipocyte and cancer cell cross-talk as a metabolic adaptation for tumor progression, there is significant interest in disrupting lipid transfer between adipocytes and cancer cells.

Leveraging our model to visualize lipid droplets in adipocytes, we became interested in identifying novel compounds that remodel adipocyte lipid droplets through lipolysis. In mammalian systems, the most commonly used cell line to study lipolysis is 3T3-L1 cells, which can be differentiated in vitro to resemble adipocytes (*Zebisch et al., 2012*). We first used the 3T3-L1 system to rapidly identify lipolysis inhibitors at high throughput and then test those hits using our zebrafish lipid droplet reporter. We reasoned that compounds that inhibit lipolysis in vitro would cause an increase in the size of the lipid droplets in vivo. To achieve this, we differentiated mouse 3T3-L1 fibroblast cells into adipocytes and conducted a chemical screen for compounds that inhibit lipolysis (*Figure 5A*), measured by quantifying glycerol in the media, a gold standard readout of lipolysis in this system (*Hellmér et al., 1989*). As a positive control, we used Atglistatin, an inhibitor of adipose triglyceride lipase (ATGL) that is known to be the rate-limiting step of lipolysis and has been shown to inhibit lipolysis in cell lines and mouse models (*Mayer et al., 2013*; *Schweiger et al., 2017*). We confirmed that Atglistatin potently inhibits lipolysis in 3T3-L1 adipocytes (*Figure 5B*). We then screened through a library of 1280 compounds of diverse chemical structures to find novel inhibitors of

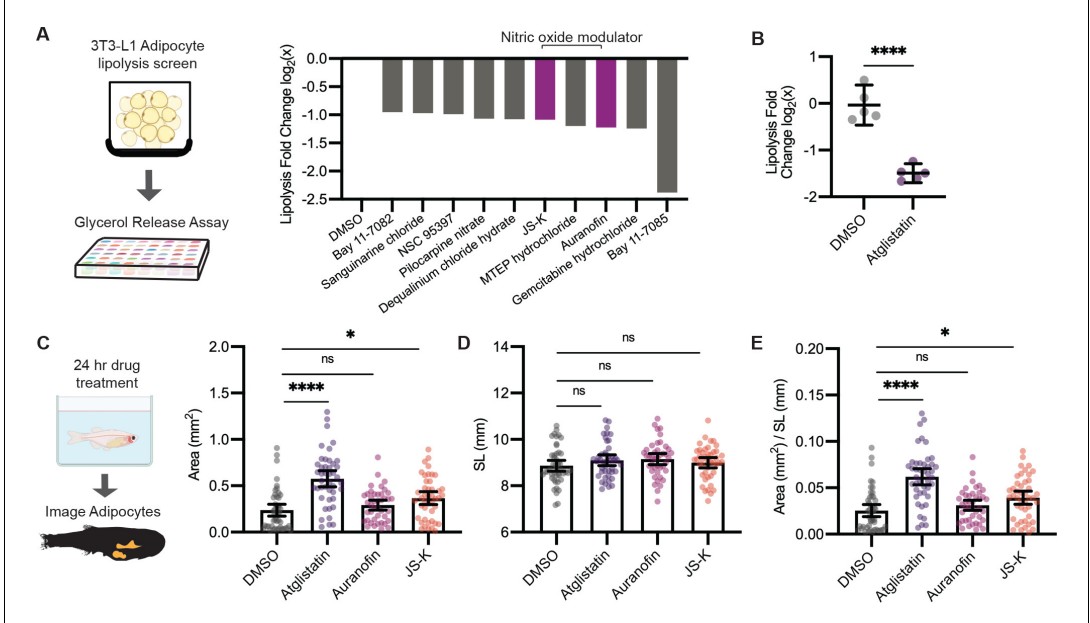

**Figure 5.** A screen to discover novel compounds that modulate lipolysis and lipid droplets in vivo. (**A**) Schematic of pharmacologic lipolysis screen in 3T3-L1 adipocytes using a glycerol release assay. Normalized $\log_2$ transformed values for top 10 drugs that inhibit lipolysis are shown. Magenta indicates compounds that modulate nitric oxide. (**B**) Normalized $\log_2$ transformed values for lipolysis inhibition in 3T3-L1 adipocytes using either dimethyl sulfoxide (DMSO) or 100 µM Atglistatin. N = 5 independent experiments. Bars indicate mean and SEM. Significance calculated via Welch's t-test; ****p<0.0001. (**C**) Schematic of experimental set-up for drug treatment. 21 days post-fertilization (dpf) *Tg(-3.5ubb:plin2-tdTomato)* zebrafish were individually placed in six-well plates with either DMSO, 40 µM Atglistatin, 1 µM Auranofin, or 1 µM JS-K for 24 hr. Adipose tissue was imaged and analyzed for (**C**) area, (**D**) standard length, and (**E**) area/standard length. Data points indicate individual fish for N = 4 independent experiments; DMSO n = 47; Atglistatin n = 44; Auranofin n = 42; JS-K n = 44. Bars indicate mean and SEM. Significance calculated via Kruskal-Wallis with Dunn's multiple comparisons test; *p<0.05, ****p<0.0001.

The online version of this article includes the following source data and figure supplement(s) for figure 5:

**Source data 1.** Complete list of lipolysis screen compounds and log2 transformed values.

**Source data 2.** Data values for Figure 5B.

**Source data 3.** Data values for Figure 5C-E.

**Figure supplement 1.** Atglistatin, Auranofin, and JS-K in vivo dose titration.

**Figure supplement 1—source data 1.** Data values for Figure 5-figure supplement 1A-F.

lipolysis. Overall, we found 29 out of 1280 compounds that led to at least a 40% reduction in lipolysis as measured by glycerol release into the media. Looking more closely at the top 10 hits from this screen, we noted that 2 of the top 10 hits (Auranofin and JS-K) modulated nitric oxide (*Figure 5A*). Nitric oxide can be used for post-translational modification of proteins via S-nitrosylation (*Stamler et al., 2001*). A previous study has shown that increased nitric oxide has a suppressive role on lipolysis, and Auranofin, a thioredoxin reductase inhibitor that promotes S-nitrosylation, can inhibit lipolysis in 3T3-L1 cells (*Yamada et al., 2015*). Similarly, JS-K is a nitric oxide donor purported to promote S-nitrosylation, but it has not been shown to play a role in lipolysis (*Nath et al., 2010*; *Shami et al., 2003*). Given that both of these top hits were in the same pathway, we chose these for in vivo validation.

Next, we asked whether these drugs could modulate lipid droplet size and lead to increased adiposity in zebrafish. We first established the maximal tolerated doses of Atglistatin, Auranofin, and JS-K in vivo (*Figure 5—figure supplement 1A–C*), and then tested their effects on adipose tissue in the *Tg(-3.5ubb:plin2-tdTomato)* fish (*Figure 5D–F*). As expected, treatment of juvenile zebrafish for 24 hr with Atglistatin caused a significant increase in adipose tissue area without affecting standard length or fish viability at the maximally tolerated dose. Consistent with our screen results, we found that JS-K also significantly increased adiposity (*Figure 5C,E*). These effects were specific to the adipose tissue as standard length was not affected (*Figure 5D*). These data indicate that modulators of nitric oxide can inhibit lipolysis in cell lines and lead to increased adiposity in vivo in zebrafish.

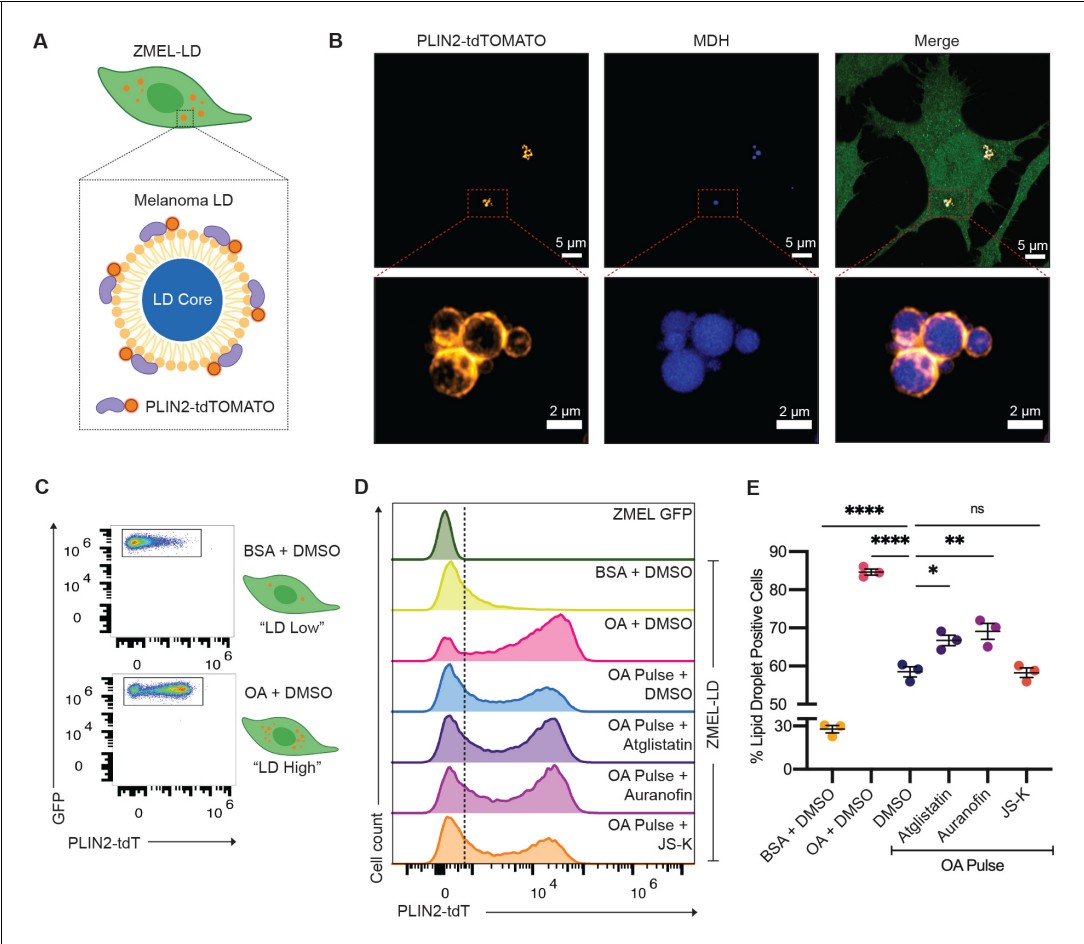

**Figure 6.** Lipolysis modulators also inhibit lipid droplet loss in melanoma cells. (**A**) Schematic of zebrafish melanoma cell line with lipid droplet reporter (ZMEL-LD) with lipid droplet labeled by PLIN2-tdTOMATO. (**B**) Confocal images of ZMEL-LD cells after 24 hr of oleic acid treatment. Panels show fluorescence signals in PLIN2-tdTOMATO, MDH (lipid droplet dye) staining, and merge of images with cytoplasmic GFP. Red box indicates position of higher magnification image of lipid droplets. (**C**) ZMEL-LD cells were treated with either bovine serum albumin (BSA) or oleic acid with dimethyl sulfoxide (DMSO) for 72 hr and then analyzed by Fluorescence Activated Cell Sorting (FACS) for PLIN2-tdTOMATO expression. (**D**) Representative histogram of PLIN2-tdTOMATO expression of ZMEL-GFP and GFP+ZMEL-LD cells with indicated drugs. Dashed line shows threshold for PLIN2-tdTOMATO expression. (**E**) Quantification of percent of GFP+ZMEL-LD cells with lipid droplets. Lipid droplet low and high controls were ZMEL-LD cells treated with BSA or oleic acid for 72 hr. For drug treatments, ZMEL-LD cells were pulsed with oleic acid for 24 hr and then given DMSO, 40 μM Atglistatin, 0.5 μM Auranofin, or 0.5 μM JS-K for 48 hr. N = 3 independent experiments. Bars indicate mean and SEM. Significance calculated via one-way ANOVA with Dunnett's multiple comparisons test; *p<0.05, ***p<0.001, ****p<0.0001.

The online version of this article includes the following figure supplement(s) for figure 6:

**Figure supplement 1.** Lipid droplet production is equivalent between ZMEL-GFP and ZMEL-LD.

Moreover, this approach demonstrates the power of this system to dissect the relationship between novel modulators of lipolysis (i.e., nitric oxide) and adiposity in vivo.

## Lipolysis modulators also inhibit lipid droplet loss in melanoma cells

Upon uptake of adipocyte-derived lipids, cancer cells can store excess lipids in lipid droplets (*Lengyel et al., 2018*). Accumulation of lipid droplets in melanoma cells has been associated with increased metastatic potential and worse clinical outcomes (*Fujimoto et al., 2020*; *Zhang et al., 2018*). The mechanisms regulating subsequent lipolysis from the lipid droplets in cancer cells are not well understood, but we reasoned that some of the same mechanisms (i.e., ATGL, nitric oxide) used in adipocytes might also be used in cancer cells. To test this, we created a stable zebrafish melanoma cell line (ZMEL) that expressed the *-3.5ubb:plin2-tdTomato* construct (*Heilmann et al., 2015*) to generate the ZMEL-LD (lipid droplet) reporter cell line (*Figure 6A*). Because melanoma cells at

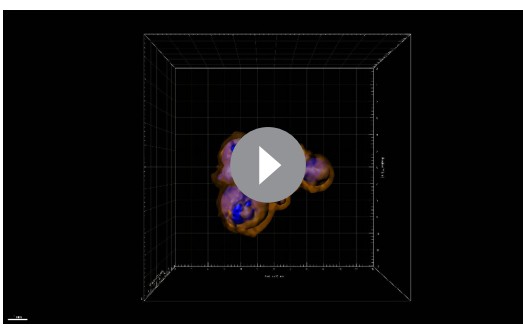

**Video 1.** 3D reconstruction of ZMEL-LD lipid droplet. ZMEL-LD cells were given oleic acid for 24 hr, fixed, and stained with the lipid droplet dye MDH. This video is a 3D reconstruction of 37 planes covering a 6 μm stack of a lipid droplet cluster in a ZMEL-LD cell. PLIN2-tdTOMATO (orange) is located outside of the lipid droplet core (blue).

https://elifesciences.org/articles/64744#video1

baseline only have few small lipid droplets, we induced their formation via extrinsic addition of oleic acid, a key fatty acid that can be transferred from the adipocyte to the melanoma cell (*Zhang et al., 2018*). We found that after a pulse of oleic acid for 24 hr, we could easily detect PLIN2-tdTOMATO expression surrounding lipid droplets marked by the lipid droplet dye Monodansylpentane (MDH) (*Figure 6B*, *Video 1*), similar to what we saw in the adipocytes (*Figure 1*). Similar to what we did in the whole fish, we wanted to be sure that the PLIN2-tdTOMATO transgene itself did not alter lipid droplets, so we compared lipid droplet formation in ZMEL-LD and parental cell line (ZMEL-GFP) via imaging and flow cytometry (*Figure 6—figure supplement 1A,B*). Using the lipid droplet dye Lipidtox as a proxy for lipid droplet content, we pulsed the cells with increasing amounts of oleic acid and found no difference in Lipidtox median fluorescence intensity (MFI) between the two cell lines (*Figure 6—figure supplement 1B*), indicating that the transgene was not having any unexpected effect. Next, we assessed the sensitivity of the PLIN2-tdTOMATO transgene versus Lipidtox in reporting changes in lipid droplets of ZMEL-LD. While both PLIN2-tdTOMATO and Lipidtox expression increased accordingly with oleic acid, PLIN2-tdTOMATO expression provided a much greater dynamic range (*Figure 6—figure supplement 1C–F*), highlighting its advantages over standard lipid dyes like Lipidtox.

To validate whether the lipolysis-inhibiting compounds we identified above could modulate lipid droplets in ZMEL-LD cells, we utilized the same flow cytometry assay to measure PLIN2-tdTOMATO expression. We treated ZMEL-LD cells for 72 hr with either BSA or oleic acid as controls for low- or high-lipid droplet cell populations (*Figure 6C*). We then tested the effects of the lipolysis inhibitors Atglistatin, Auranofin, and JS-K. We pulsed the ZMEL-LD cells with oleic acid for 24 hr (to induce lipid droplets) and then measured the subsequent decay in signal over the ensuing 48 hr, which is expected to decrease due to gradual lipolysis of the lipid droplets. Compared to cells with oleic acid pulse and DMSO, cells given JS-K did not differ in the percent of lipid droplet-positive cells (*Figure 6D,E*). In contrast, cells treated with Atglistatin and Auranofin demonstrated significantly higher lipid droplet-positive cells (*Figure 6D,E*). These data indicate that similar to adipocytes, ATGL is a key regulatory step in lipolysis in melanoma cells. Moreover, we found that nitric oxide, which was identified in our adipocyte screen, is similarly a modulator of lipolysis in the melanoma context and can be utilized for future studies to target adipocyte-melanoma cell cross-talk. We do not yet understand why different nitric oxide donors are more or less potent in adipocytes (where JS-K is a better inhibitor in vivo) versus melanoma cells (where Auranofin is a better inhibitor), but this could reflect differences in pharmacokinetics or cell-type-specific lipid droplet regulation.

## Discussion

Lipid droplets are cytosolic storage organelles for cellular lipids, which are dynamically regulated in response to metabolic and oxidative perturbations (*Jarc and Petan, 2019*). Changes in the lipid droplet content of a cell can occur in response to a variety of factors, including hypoxia, reactive oxygen stress, ER stress, and alterations in nutrient availability (*Bailey et al., 2015*; *Bensaad et al., 2014*; *Chitraju et al., 2017*; *Velázquez et al., 2016*; *Vevea et al., 2015*; *Cabodevilla et al., 2013*; *Nguyen et al., 2017*). However, the regulatory mechanisms driving these processes remain incompletely understood. Furthermore, lipid droplets are highly heterogeneous, and the pathways that regulate lipid droplet dynamics in specific cell types warrant investigation.

To address such questions, we report the first lipid droplet reporter in a vertebrate model organism. We show that our *plin2-tdtomato* reporter faithfully marks the lipid droplet in vivo. As a further

validation of our approach, while our manuscript was in review, a similar approach using knockin at the endogenous *plin2* promoter was recently described (*Wilson et al., 2021*). The combination of our reporter with the in vivo system of the *casper* zebrafish enables flexible and robust imaging approaches to examine lipid droplet regulation and function. In particular, the ease of chemical and genetic manipulation of the zebrafish combined with high-throughput imaging approaches enables interrogation of relevant pathways in a cell type-specific manner. Furthermore, the capacity for intra-vital imaging creates the opportunity to conduct longitudinal analysis of lipid droplet dynamics across developmental time and in disease contexts between single animals.

Here, we demonstrate the capabilities of the *Tg(-3.5ubb:plin2-tdTomato)* line by taking advantage of the fact that white adipocytes are readily labeled by PLIN2-tdTOMATO expression. This labeling enables the study of individual adipocytes and adipose tissue in adult and juvenile zebrafish. We show that the *Tg(-3.5ubb:plin2-tdTomato)* reporter can be used as an alternative to lipophilic fluorescent dyes to study adipose tissue across zebrafish developmental stages while preserving normal adipose tissue development.

We establish the utility of this transgenic line to study the regulation of adipose tissue by both diet and pharmacologic perturbations. We focused on visceral adipose tissue due to its role as an endocrine organ and the central regulator of organismal metabolism (*Fox et al., 2007*; *Le Jemtel et al., 2018*; *Verboven et al., 2018*). We show that our *Tg(-3.5ubb:plin2-tdTomato)* line is sensitive enough to capture quantitative changes in visceral adipose tissue after short-term pharmacologic treatments with known regulators of adipocyte lipolysis. Furthermore, by combining top hits from a large-scale in vitro chemical screen in 3T3-L1 adipocytes with our reporter we uncovered a novel role for nitric oxide in modulating adipocyte lipolysis and adipose tissue dynamics.

Beyond pharmacologic perturbations, we also demonstrate the power of this line to perform longitudinal analyses of diet-induced perturbations on adipose tissue area across multiple time points. This work yielded compelling insights into the dynamics of adipose tissue response to both prolonged fasting and high-fat diet. Differences in the kinetics of each response suggest a complex relationship between adipose tissue development and the nutritional and energetic requirements of the developing organism, which merits future investigation. Collectively, these data illustrate the potential of our model to yield novel insights into the regulation of visceral adipose tissue, including the context of obesity.

While our studies show that this tool can be used to increase our understanding of adipocyte biology, it can also be utilized to study lipid droplets in other contexts as well. Lipid droplets are ubiquitous across almost all cell types. Therefore, this model could be applied to study the regulation of lipid droplets in the development and function of other adipose depots and additional cell types, such as muscle and hepatocytes (*Bosma, 2016*; *Wang et al., 2013*). In the disease context, we focused on the role of lipid droplets in cancer, since tumor cells can take up lipids from adipocytes and then package them into lipid droplets (*Balaban et al., 2017*; *Lengyel et al., 2018*; *Nieman et al., 2011*; *Zhang et al., 2018*). This transfer of lipids has been linked to disease progression, making the regulation of lipid release from the lipid droplet through subsequent lipolysis in the tumor cell of particular interest. We found that regulation of lipolysis by ATGL and nitric oxide pathways is conserved between adipocytes and melanoma cells although phenotypes downstream of nitric oxide may be cell type specific. Collectively, this underscores the complexity of lipid droplet regulation and emphasizes the importance of studying these processes in multiple cell types. We believe that our model will serve as a powerful tool to study cell type-specific regulation of lipid droplet biogenesis and function while preserving the endogenous structural and metabolic environment of an in vivo system.

## Materials and methods

### Key resources table

| Reagent type (species) or resource | Designation | Source or reference | Identifiers | Additional information |
|---|---|---|---|---|
| Cell line (*Danio rerio*) | ZMEL | *Heilmann et al., 2015* | | mitfa:BRAFV600E/p53-/- |

*Continued on next page*

*Continued*

| Reagent type (species) or resource | Designation | Source or reference | Identifiers | Additional information |
|---|---|---|---|---|
| Cell line (*Danio rerio*) | ZMEL-LD | This paper | (-3.5ubb:plin2-tdtomato) in pDestTol2pA2-blastocidin | |
| Cell line (*Mus musculus*) | 3T3-L1 | ZenBio | SP-L1-F | |
| Strain, strain background (*Danio rerio*) | *casper* | *White et al., 2008* | mitfaw2/w2;mpv17a9/a9 | |
| Strain, strain background (*Danio rerio*) | (-3.5ubb:plin2-tdtomato) | This paper | | |
| Recombinant DNA reagent | (-3.5ubb:plin2-tdtomato) in pDestTol2CG2 (plasmid) | This paper | | |
| Recombinant DNA reagent | (-3.5ubb:plin2-tdtomato) in pDestTol2pA2-blastocidin (plasmid) | This paper | | |
| Recombinant DNA reagent | pDestTol2pA2-blastocidin (plasmid) | *Heilmann et al., 2015* | | |
| Recombinant DNA reagent | pDestTol2CG2 (plasmid) | *Kwan et al., 2007* | | |
| Sequence-based reagent | PLIN2 cDNA FWD | This paper | PCR primer | AAAGCAGGCTCCACCATGAGCTTTCTTCTGTACTTGAAACTG |
| Sequence-based reagent | PLIN2 cDNA REV | This paper | PCR primer | GCCCTTGCTCACCATTTCAGTGACTTGAAGGGTCCTCTGT |
| Sequence-based reagent | PLIN2-TMT FWD | This paper | PCR primer | GCCGCCCCCTTCACCATGAGCTTTCTTCTGTACTTGAAAC |
| Sequence-based reagent | PLIN2-TMT REV | This paper | PCR primer | GCCCTTGCTCACCATTTCAGTGACTTG |
| Sequence-based reagent | tdTOMATO ME PLIN2 FWD | This paper | PCR primer | ATGGTGAGCAAGGGCGAG |
| Sequence-based reagent | tdTOMATO ME PLIN2 REV | This paper | PCR primer | GGTGAAGGGGGCGGC |
| Commercial assay or kit | Cloneamp HiFi PCR Premix | Takara | Catalog #639298 | |
| Commercial assay or kit | In-Fusion HD Cloning Plus | Takara | Catalog #638920 | |
| Commercial assay or kit | Gateway LR Clonase Enzyme mix | Thermo Fisher | Catalog #11791019 | |
| Commercial assay or kit | Zymogen Quick RNA Miniprep Kit | Zymo Research | Catalog #R1054 | |
| Commercial assay or kit | Invitrogen SuperScriptIII First-Strand Synthesis SuperMix Kit | Thermo Fisher | Catalog #18080400 | |
| Commercial assay or kit | NucleoSpin Gel and PCR Clean up | Takara | Catalog #740609.50 | |
| Commercial assay or kit | Free Glycerol Reagent | Sigma-Aldrich | Catalog #F6428 | |
| Chemical compound | Glycerol Standard Solution | Sigma-Aldrich | Catalog #G7793 | |
| Other | HCS LipidTOX deep red | Thermo Fisher | Catalog #H34477 | 1:250 or 1:500 |
| Other | BODIPY 493/503 | Thermo Fisher | Catalog #D3922 | 5 or 10 ng/µl |
| Other | AUTODOT Visualization Dye (MDH) | Abcepta | Catalog #SM1000a | 1:500 |
| Chemical compound, drug | Forskolin | Sigma-Aldrich | Catalog #F6886 | 5 µM (fish) |
| Chemical compound, drug | Auranofin | Sigma-Aldrich | Catalog #A6733 | 1 µM (fish), 0.5 µM (cells) |
| Chemical compound, drug | JS-K | Sigma-Aldrich | Catalog #J4137 | 1 µM (fish), 0.5 µM (cells) |
| Chemical compound, drug | Atglistatin | Sigma-Aldrich | Catalog #SML1075 | 40 µM |
| Chemical compound | Oleic Acid-Albumin | Sigma-Aldrich | Catalog #O3008-5M | |

*Continued on next page*

*Continued*

| Reagent type (species) or resource | Designation | Source or reference | Identifiers | Additional information |
|---|---|---|---|---|
| Chemical compound | LOPAC 1280 Library | Sigma-Aldrich | Catalog #LO1280 | |
| Antibody | Anti-RFP antibody (rabbit polyclonal) | Rockland | Catalog #600-401-379, RRID:AB_2209751 | (1:500), (1 µL) |
| Software, algorithm | MATLAB | Mathworks | | |
| Software, algorithm | PRISM | Graphpad | | |
| Software, algorithm | FIJI | Schindelin et al., 2012 | | |
| Software, algorithm | FlowJo | Becton, Dickinson and Company | | |
| Other | High-fat diet | Sparos | | See detailed description of high-fat diet contents in the 'Methods' section below |

## Cloning of -3.5ubb:plin2-tdtomato

To clone the *plin2* cDNA, tissue from the muscle and heart of adult *casper* zebrafish was dissected, pooled, and then RNA was isolated using the Zymogen Quick RNA Miniprep Kit (Zymo Research, Irvine, USA; catalog #R1054) according to manufacturer's instructions. The Invitrogen SuperScriptIII First-Strand Synthesis SuperMix Kit (Thermo Fisher, Waltham, USA; catalog #18080400) was used according to manufacturer's instructions to produce cDNA. CloneAmp HiFi PCR Premix (Takara, Mountain View, USA; catalog #639298) was used to PCR amplify the PLIN2 cDNA and gel purified via NucleoSpin Gel and PCR Clean Up (Takara, Mountain View, USA; catalog #740609.50). To generate pME-PLIN2-tdTOMATO, the PLIN2 cDNA was inserted on the 5' end of pME-tdTOMATO using In-Fusion HD Cloning Plus (Takara, Mountain View, USA; catalog #638920). Gateway cloning using the Gateway LR Clonase Enzyme mix (Thermo Fisher, Waltham, USA; catalog #11791019) was employed to create the *-3.5ubb:plin2-tdTomato* construct with p5E-ubb, pME-PLIN2-tdTOMATO, and p3E-polyA into pDestTol2pA2-blastocidin (cells) (*Heilmann et al., 2015*) or pDestTol2CG2 (zebrafish) (*Kwan et al., 2007*).

## Zebrafish husbandry

All zebrafish experiments were carried out in accordance with institutional animal protocols. All zebrafish were housed in a temperature- (28.5°C) and light-controlled (14 hr on, 10 hr off) room. Fish were initially housed at a density of five fish per liter and fed three times per day using rotifers and pelleted zebrafish food. Anesthesia was done using Tricaine (Western Chemical Incorporated, Ferndale, USA) with a stock of 4 g/l (protected for light) and diluted until the fish was immobilized. All procedures were approved by and adhered to Institutional Animal Care and Use Committee (IACUC) protocol #12-05-008 through Memorial Sloan Kettering Cancer Center.

## Generation of *Tg(-3.5ubb:plin2-tdTomato)*

The *ubb:plin2-tdTomato* plasmid was injected into *casper* embryos with Tol2 mRNA to introduce stable integration of the *ubb:plin2-tdTomato* cassette. Fish with GFP+ hearts (due to pDestTol2CG) were selected and outcrossed with *casper* fish to produce the F1 generation. F1 zebrafish with GFP + hearts and validated PLIN2-tdTOMATO-expressing adipocytes were outcrossed to generate F2 and F3 generation zebrafish for experiments.

## Zebrafish imaging and analysis

Zebrafish were imaged using an upright Zeiss AxioZoom V16 Fluorescence Stereo Zoom Microscope with a x0.5 (for adult fish) or x1.0 (for juvenile fish) adjustable objective lens equipped with a motorized stage, brightfield, and Cy5, GFP, and tdTomato filter sets. To acquire images, zebrafish were lightly anesthetized with 0.2% Tricaine. Images were acquired with the Zeiss Zen Pro v2 and exported as CZI files for visualization using FIJI or analysis using FIJI (to manually quantify standard length) and MATLAB (Mathworks, Natick, USA).

Our adipocyte segmentation approach utilized the Image Processing Toolbox within MATLAB. Because the zebrafish gut is highly autofluorescent, we chose a threshold for the GFP channel to categorize as background signal and subtracted it from a determined threshold for the tdTOMATO channel. We used a set size to crop images around the tdTOMATO-positive signal and created a mask for the adipose tissue. Within the masked area, we applied a higher tdTOMATO threshold to segment the fluorescent signal from the adipocytes. Finally, we quantified the number of pixels above the threshold to quantify adipose tissue area. MATLAB code is available for download at https://github.com/dlumaquin/PLIN2-tdT-Adipo-Quant.git (copy archived at URL to be added).

To quantify BODIPY-stained adipose tissue, we utilized FIJI to autothreshold GFP signal. We removed background autofluorescence by subtracting Cy5 autothresholded signal. We used the polygon tool to outline and quantify the resulting segmented adipose area.

For visualization purposes, the segmented images were color filtered on Adobe Photoshop from grayscale to gold color scale.

## BODIPY staining of zebrafish

Adult zebrafish were placed in tanks with 10 ng/µl BODIPY 493/503 (Thermo Fisher, Waltham, USA; catalog #D3922) for 30 min in the dark. Fish were washed and then placed in new tanks with fresh water for 2 hr. Fish were washed again to remove any residual BODIPY, then anesthetized and imaged as indicated above for whole adipose tissue.

Higher resolution images of zebrafish adipocytes were acquired using the Zeiss LSM 880 inverted confocal microscope using a x10 objective. Zebrafish were lightly anesthetized with 0.2% Tricaine and mounted on a glass bottom dish (MatTek, Ashland, USA; catalog #P35G-1.5–20 C) with 0.1% low gelling agarose (Sigma-Aldrich, St. Louis, USA; catalog #A9045-25G).

## Staining to track adipose depot development

For BODIPY staining, 5, 14, and 21 dpf fish were incubated with 5 ng/µl BODIPY 493/503 in dishes for 30 min in the dark. Fish were washed with fresh E3 every 45 min for 1.5 hr, then anesthetized and imaged as indicated above for whole adipose tissue. 42 dpf and adult zebrafish were stained in p1000 tip boxes for 30 or 15 min, respectively, in the dark. Fish were then placed back on system to wash with fresh water for 1.5 hr and then anesthetized and imaged as indicated below for whole adipose tissue.

## Image analysis to track adipose depot development

To track adipose depot development across multiple stages, F2 (-3.5ubb:plin2-tdTomato) zebrafish were outcrossed with *caspers* to generate F3 fish expressing the transgene and control siblings. Control and *Tg(-3.5ubb:plin2-tdTomato)* fish were raised to a standard density of 25 fish per 2.8 l tank. At the appropriate time points, 5, 14, 21, and 42 dpf or adult-stage zebrafish were removed from the tank and stained for BODIPY as described above.

Images were acquired in three segments per fish along the anterior to posterior axis to capture the entire body of the fish. Adipose depots were classified based on the previously described system (*Minchin and Rawls, 2017a*). To determine the presence of each adipose depot, images were thresholded in FIJI for GFP (BODIPY493/503) or tdTomato (-3.5ubb:plin2-tdTomato) using control fish as the reference. A depot was scored as present based on positive signal corresponding to the presence of adipocytes for at least one sub-depot in the appropriate anatomic location.

## IHC for tdTOMATO

Zebrafish were sacrificed in an ice bath for at least 15 min. For adults, zebrafish tails were dissected. For juvenile zebrafish, the entire fish was used for fixation. Selected zebrafish were fixed in 4% paraformaldehyde for 72 hr at 4°C, washed in 70% ethanol for 24 hr, and then paraffin embedded. Fish were sectioned at 5 µm and placed on Apex Adhesive slides, baked at 60°C, and then stained with antibodies against tdTomato (1:500, Rockland, #600-401-379). Histology was performed and stained by Histowiz.

## Zebrafish fast

*Tg(-3.5ubb:plin2-tdTomato)* F2 fish were outcrossed with *caspers* to generate the F3 generation. F3 fish were raised at a standard density of 25 fish per 2.8 l tank. For BODIPY staining experiments, 21 dpf fish were separated into new tanks that received standard feed or were fasted for 7 days. Prior to imaging, food was withheld for 3–6 hr to clear the gut. Fish were anesthetized with Tricaine and imaged as described above to quantify BODIPY-stained visceral adipose tissue area and standard length. For the time course fast, 21 dpf *Tg(-3.5ubb:plin2-tdTomato)* fish were separated into new tanks that received standard feed or fasted. Prior to imaging, food was withheld for 3–6 hr to clear the gut. Fish were anesthetized with Tricaine and imaged on days 0, 2, 5, and 7 to quantify PLIN2-tdTOMATO-positive visceral adipose tissue area and standard length.

## High-fat diet feeding

*Tg(-3.5ubb:plin2-tdTomato)* F3 zebrafish were raised at a standard density of 25 fish per 2.8 l tank. At 21 dpf, the zebrafish were placed in 0.8 l tanks and fed either a high-fat or control diet (Sparos, Portugal) for up to 14 days. Fish were then imaged for PLIN2-tdTOMATO expression at days 0, 7, and 14 after the start of diet. Prior to imaging, fish were put in a new tank and food withheld for ~16 hr. Zebrafish were at equal density for control and experimental groups, ranging from 15 to 30 fish per tank. Fish were fed 0.1 g feed per tank per day split over two feedings. The high-fat and control diets were customized and produced at Sparos Lda (Olhão, Portugal), where powder ingredients were initially mixed according to each target formulation in a double-helix mixer, and thereafter ground twice in a micropulverizer hammer mill (SH1; Hosokawa-Alpine, Germany). The oil fraction of the formulation was subsequently added and diets were humidified and agglomerated through low-shear extrusion (Dominioni Group, Italy). Upon extrusion, diets were dried in a convection oven (OP 750-UF; LTE Scientifics, United Kingdom) for 4 hr at 60°C, and subsequently crumbled (Neuero Farm, Germany) and sieved to 400 μm. Experimental diets were analyzed for proximal composition. The Sparos control diet contains 30% fishmeal, 33% squid meal, 5% fish gelatin, 5.5% wheat gluten, 12% cellulose, 2.5% soybean oil, 2.5% rapeseed oil, 2% vitamins and minerals, 0.1% vitamin E, 0.4% antioxidant, 2% monocalcium phosphate, and 2.2% calcium silicate. The Sparos HFD contains 30% fishmeal, 33% squid meal, 5% fish gelatin, 5.5% wheat gluten, 12% palm oil, 2.5% soybean oil, 2.5% rapeseed oil, 2% vitamins and minerals, 0.1% vitamin E, 0.4% antioxidant, 2% monocalcium phosphate, and 2.2% calcium silicate.

## 3T3-L1 cell culture

3T3-L1 cells were acquired from ZenBio and their differentiation protocol was followed. Cells were received at passage 8 and split to a maximum of passage 12 as per the recommendations of the company. 96-well plates were coated with fibronectin (EMD Millipore, Burlington, USA; catalog #FC010) diluted 1:100 in phosphate-buffered saline (PBS) for at least 30 min to promote improved adherence of cells to the dish. 3T3-L1 cells were first cultured in PM-1-L1 Preadipocyte Medium and allowed to grow to 100% confluence. PM-1-L1 medium was changed every 48–72 hr. 48 hr after reaching 100% confluence, cells were changed to DM-1-L1 Differentiation Medium for 72 hr and then changed to AM-1-L1 Adipocyte Medium. AM-1-L1 Adipocyte Medium was changed every 48–72 hr. Once in AM-1-L1, the medium was changed gently with a multichannel pipette, and only 150 μl of the 200 μl was replaced to prevent touching the bottom of the well with the pipette tip. After 2–3 weeks in AM-1-L1, the 3T3-L1 developed significantly large lipid droplets and were used in the screen.

## LOPAC library screen

The LOPAC library includes 1280 clinically relevant compounds with annotated targets or pathways. The workflow of the screen involved drug or vehicle control of the 3T3-L1 adipocytes for 24 hr in serum-free media. After 24 hr, 100 μl of the media supernatant was collected to measure secreted glycerol using the Free Glycerol Reagent (Sigma-Aldrich, St. Louis, USA; catalog F6428) following the associated glycerol assay protocol.

The medium (screen media) used for drug treatment was phenol-free DMEM supplemented with 0.2% BSA FFA-free (Sigma-Aldrich, St. Louis, USA; catalog 9048-46-8). The 1280 compounds were aliquoted as 2 μl at 1 mM into 16 × 96-well plates and stored at −20°C. Upon thawing, 198 μl of

screen media was added to the well, bringing the final drug concentration for all compounds in the screen to 10 µM. Control vehicle was 1% DMSO served as a negative control and 1 uM isoproterenol served as a positive control in the screen. This medium containing LOPAC drugs, DMSO, and isoproterenol was transferred to 3T3-L1 cells and incubated for 24 hr.

To measure glycerol release as a readout for lipolysis, 100 µl of Free Glycerol Reagent was aliquoted per well of a 96-well plate. 10 µl of supernatant media from 3T3-L1 adipocytes was then added to each well. A standard curve was produced by using Glycerol Standard Solution (Sigma-Aldrich, St. Louis, USA; catalog G7793). The plate was incubated at 37°C for 5 min and then developed with a plate reader set to detect absorbance at 540 nm. Using the standard curve, a fit equation was developed in Excel to convert the absorbance values into glycerol concentrations. To take into account differences that occur in wells on the edge versus middle of the plate, all well positions across all plates in the screen were averaged to create a normalization factor for any given position on the plate. These normalized values were then used to determine top hits for compounds that block lipolysis.

### Glycerol release assay with Atglistatin

3T3-L1s were differentiated on a fibronectin-coated 96-well dish. At the start of the lipolysis experiment, 3T3-L1s were changed to serum-free DMEM supplemented with 0.2% BSA FFA-free (Sigma-Aldrich, St. Louis, USA; catalog 9048-46-8). The medium was supplemented with 1% DMSO for negative control or 1 uM isoproterenol to induce lipolysis or ±100 µM Atglistatin (Sigma-Aldrich, St. Louis, USA; catalog SML1075) to block lipolysis and cells were incubated for 24 hr.

To measure glycerol release, 100 µl of Free Glycerol Reagent was aliquoted per well of a new 96-well plate. 10 µl of supernatant media from 3T3-L1 adipocytes was then added to each well. A standard curve was produced by using Glycerol Standard Solution (Sigma-Aldrich, St. Louis, USA; catalog G7793). The plate was incubated at 37°C for 5 min and then developed with a plate reader set to detect absorbance at 540 nm. Using the standard curve, a fit equation was developed in Excel to convert the absorbance values into glycerol concentrations.

### Zebrafish drug treatments

*Tg(-3.5ubb:plin2-tdTomato)* zebrafish were outcrossed with *caspers* to generate the F2 or F3 generation. F2 or F3 fish were raised at a standard density of 50 fish per 6.0 l tank. For drug treatment, fish were removed from the system at 21 dpf and placed at a density of one fish per well in a six-well plate with 10 ml of E3 per well. To evaluate viability, fish were treated for 24 hr and quantified for live and dead larvae. After a 24 hr incubation with the drug, fish were anesthetized with Tricaine and imaged using the described protocol to quantify (1) standard length and (2) area of PLIN2-tdTOMATO expression corresponding to visceral adipose tissue area. Fish were treated with Forskolin (Sigma-Aldrich, St. Louis, USA; catalog #F6886), Auranofin (Sigma-Aldrich, St. Louis, USA; catalog #A6733), JS-K (Sigma-Aldrich, St. Louis, USA; catalog #J4137), or Atglistatin (Sigma-Aldrich, St. Louis, USA; catalog #SML1075[1]), which were all dissolved in DMSO.

### Generation of ZMEL-LD cell line

The ZMEL zebrafish melanoma cell line was derived from a tumor of a *mitfa:BRAF$^{V600E}$/p53$^{-/-}$* zebrafish as described previously (*Heilmann et al., 2015*). ZMEL cells constitutively express eGFP driven by the *mitfa* promoter (*Heilmann et al., 2015*). ZMEL cells were grown at 28°C in a humidified incubator in DMEM (Gibco, Waltham, USA; catalog #11965) supplemented with 10% fetal bovine serum (FBS) (Gemini Bio, #100–500), 1x penicillin/streptomycin/glutamine (Gibco, Waltham, USA; catalog #10378016), and 1x GlutaMAX (Gibco, Waltham, USA; catalog #35050061). To generate the ZMEL-LD cells, ZMEL cells were nucleofected with the *ubb:plin2-tdtomato* plasmid using the Neon Transfection System (Thermo Fisher, Waltham, USA; catalog #MPK10096), selected for 2 weeks in blasticidine-supplemented media at 4 µg/µl (Sigma-Aldrich, St. Louis, USA; catalog #15205–25 MG), and FACS sorted for GFP and tdTOMATO double-positive cells. ZMEL and ZMEL-LD cells underwent routine *Mycoplasma* testing, most recently in January 2021.

## ZMEL-GFP and ZMEL-LD imaging

Eight-well Nunc Lab-Tek Chambered Coverglass was coated with 1:100 dilution of fibronectin in Dulbecco's Phosphate Buffered Saline (DPBS) (Millipore Sigma, Burlington, USA; catalog #FC010-5MG) for 30 min and then washed with DPBS (Thermo Scientific, Waltham, USA; catalog #14190–250). ZMEL-GFP or ZMEL-LD cells were seeded at 30,000 cells per well and left to adhere for 24 hr. A medium supplemented with oleic acid (Sigma-Aldrich, St. Louis, USA; catalog #O3008-5ML) was added for 24 hr. Cells were fixed with 2% paraformaldehyde (Santa Cruz Biotechnology, Santa Cruz, USA; catalog #sc-281692) for 45 min and washed with DPBS. For MDH staining, cells were permeabilized with 0.1% Triton-X (Thermo Fisher, Waltham, USA; catalog #PI85111) for 30 min at room temperature, washed, and stained with 1:500 MDH (Abcepta, San Diego, USA; catalog #SM1000a) for 15 min. Cells were imaged on the Zeiss LSM 880 inverted confocal microscope with AiryScan using a x63 oil immersion objective. For Lipid staining, cells were stained with 1:500 Lipidtox Deep Red (Thermo Fisher, Waltham, USA; catalog #H34477) and 1:2000 Hoechst 33342 (Thermo Fisher, Waltham, USA; catalog #H3570) for 30 min. Cells were imaged on the Zeiss LSM 880 inverted confocal microscope using a x40 oil immersion objective. Confocal stacks were visualized via FIJI, and 3D reconstruction was created using Imaris (Bitplane Inc, Concord, USA).

## ZMEL-LD FACS analysis

ZMEL Dark (no fluorescence), ZMEL-GFP, and ZMEL-LD cells were plated on fibronectin-coated six-well plates at a density of 500,000 cells in 1 ml of media per well. At 24 hr after plating, cells were given either 150 µM of BSA or oleic acid with 1 µl of DMSO. At 48 and 72 hr after plating, lipid droplet low and high controls were switched to fresh media with 150 µM of BSA or oleic acid with 1 µl of DMSO. Cells pulsed with oleic acid received fresh media with 150 µM of BSA with either 40 µM Atglistatin, 0.5 µM Auranofin, or 0.5 µM JS-K. At 96 hr after plating, cells were trypsinized, washed with DPBS, and resuspended in DMEM supplemented with 2% FBS, 1x penicillin/streptomycin/glutamine, and 1x GlutaMAX. Cells were stained for viability with 1:1000 DAPI and strained through the Falcon FACS Tube with Cell Strainer Cap (Thermo Fisher, Waltham, USA; catalog #08-771-23). For Lipidtox comparison, cells were given either BSA or indicated concentrations of oleic acid for 24 hr. Cells were trypsinized, washed with DPBS, stained with 1:250 Lipid Deep Red and 1:1000 DAPI for 10 min, and strained through the Falcon FACS Tube with Cell Strainer Cap. Data was acquired via the Beckman Coulter CytoFLEX Flow Cytometer (Beckman Coulter, Miami, USA) and analyzed using FlowJo software (BD Biosciences, San Jose, USA).

## Schematics

Schematics and illustrations were generated using Biorender on biorender.com.

## Statistics

All statistical analyses were performed using GraphPad Prism 8 (Graphpad, San Diego, USA). Data are presented as mean ± 95% confidence interval (CI) or standard error of the mean (SEM). $p < 0.05$ was considered statistically significant. Statistical tests used are noted in the figure legend. All experiments were done with at least three independent replicates. For in vivo experiments, N denotes the number of independent experiments while n denotes the number of individual fish. Imaging analyses utilized FIJI, Imaris, and MATLAB software.

## Availability of resources

All zebrafish cell lines and transgenic lines are available upon request. In addition, the *Tg(-3.5ubb: plin2-tdTomato)* zebrafish will be deposited at the Zebrafish International Resource Center.

## Acknowledgements

We thank members at the Memorial Sloan Kettering Cancer Center Aquatics Core, Molecular Cytology Core, and Flow Cytometry Core for their contributions to this work. We thank Dr. Mohita Tagore and Dr. Ting-Hsiang (Richard) Huang for comments on the project and manuscript.

## Additional information

### Competing interests
Richard M White: Senior editor, *eLife*. The other authors declare that no competing interests exist.

### Funding

| Funder | Grant reference number | Author |
|---|---|---|
| National Institutes of Health | F30 CA254152 | Dianne Lumaquin |
| National Institutes of Health | T32GM007739-42 | Dianne Lumaquin Joshua M Weiss Abderhman Abuhashem |
| National Institutes of Health | 5K00CA223016-04 | Emily Montal |
| National Institutes of Health | F30 HD 103398 | Abderhman Abuhashem |
| National Institutes of Health | F31 AR079215 | Eleanor Johns |
| National Institutes of Health | R25CA020449 | David Ola |
| Melanoma Research Alliance | | Richard M White |
| National Institutes of Health | R01CA229215 | Richard M White |
| National Institutes of Health | R01CA238317 | Richard M White |
| National Institutes of Health | DP2CA186572 | Richard M White |
| Pershing Square Foundation | | Richard M White |
| Mark Foundation For Cancer Research | | Richard M White |
| Harry J. Lloyd Charitable Trust | | Richard M White |
| National Cancer Institute | P30 CA008748 | Richard M White |
| Memorial Sloan-Kettering Cancer Center | | Richard M White |
| Consano | | Richard M White |
| SSC | | Richard M White |
| American Cancer Society | American Cancer Society Research Scholar | Richard M White |

The funders had no role in study design, data collection and interpretation, or the decision to submit the work for publication.

### Author contributions
Dianne Lumaquin, Eleanor Johns, Conceptualization, Data curation, Formal analysis, Validation, Investigation, Visualization, Methodology, Writing - original draft, Writing - review and editing; Emily Montal, Data curation, Formal analysis, Investigation, Methodology, Writing - original draft, Project administration, Writing - review and editing; Joshua M Weiss, Conceptualization, Data curation, Formal analysis, Investigation; David Ola, Data curation, Investigation; Abderhman Abuhashem, Investigation; Richard M White, Conceptualization, Supervision, Funding acquisition, Writing - original draft, Project administration, Writing - review and editing

### Author ORCIDs
Dianne Lumaquin (ID) https://orcid.org/0000-0002-5285-0467
Eleanor Johns (ID) https://orcid.org/0000-0001-6802-041X
Joshua M Weiss (ID) http://orcid.org/0000-0001-7340-4004
Richard M White (ID) https://orcid.org/0000-0001-9099-9169

## Ethics

Animal experimentation: Animal experimentation: This study was performed in strict accordance with the recommendations in the Guide for the Care and Use of Laboratory Animals of the National Institutes of Health. All of the animals were handled according to approved institutional animal care and use committee (IACUC) protocols (#12-05-008) of Memorial Sloan Kettering Cancer Center. The protocol was approved by the Committee on the Ethics of Animal Experiments of Memorial Sloan Kettering Cancer Center (Permit Number: D16-00199). Every effort was made to minimize suffering.

## Decision letter and Author response

Decision letter https://doi.org/10.7554/eLife.64744.sa1
Author response https://doi.org/10.7554/eLife.64744.sa2

## Additional files

### Supplementary files

• Transparent reporting form

### Data availability

All data generated or analyzed during this study are included in the manuscript and supporting files. Source data files have been provided for Figures 2-5 and Supplementary Figures 2, 3 and 5.

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
