## [Decision Letter]

**Acceptance summary:**

In this rigorously revised manuscript you and your colleagues show that your Pin2-tdTomato fusion protein system (in whole zebrafish and in melanoma cells) is a robust reporter of lipid mass. The carefully controlled time series, high-fat feeding, and dose-response experiments strengthen your conclusions. Your reporter line will be a major resource to those in the lipid droplet field. The explicit statements regarding transgenic line and source code availability are major steps in spreading the use of your resource.

**Decision letter after peer review:**

Thank you for submitting your article "An in vivo reporter for tracking lipid droplet dynamics in transparent zebrafish" for consideration by *eLife*. Your article has been reviewed by 3 peer reviewers, including Amnon Schlegel as the Reviewing Editor and Reviewer #1, and the evaluation has been overseen by Didier Stainier as the Senior Editor. The following individual involved in review of your submission has agreed to reveal their identity: James Minchin (Reviewer #2).

The reviewers have discussed the reviews with one another and the Reviewing Editor has drafted this decision to help you prepare a revised submission.

In this Tools and Resources manuscript, the authors present a transgenic zebrafish line ubiquitously expressing a lipid droplet coat protein gene fused to a fluorescent protein. The authors characterize the properties of this reporter in labeling adipocyte lipid droplets of optically transparent juveniles and adults casper mutants. Experiments are presented using this reporter construct to validate the results of a chemical screen conducted in in vitro-differentiated adipocytes, revealing a role for nitric oxide signaling in controlling lipolysis. Several hits from this screen are also validated in a zebrafish melanoma cell line system expressing the lipid droplet reporter system. The authors posit transgenic reporter system will be useful to workers interested in lipid droplet accumulation in an intact vertebrate animal.

The authors prepared a transgenic zebrafish whose official name should be stated as Tg-3.5ubb:plin2-tdTomato throughout the manuscript. This line expresses a fusion protein of the lipid droplet coat protein Perilipin 2 with a C-terminally fused tdTomato fluorescent reporter (Figure 1). This line was prepared to establish a system for evaluating lipid droplets in an intact vertebrate model organism, something not available to the research community prior to this study. The authors posit this tool will be of use to other researchers interested in studying lipid droplet biology with implications for several diseases.

The authors use the reporter line to examine the appearance of lipid droplet area in the visceral depots of late larvae (Figure 2); monitor the decrease lipid droplet area in starvation and following adrenergic signaling-driven lipolytic states (figure 3); and to demonstrate an increase in adipose area in a high-fat dietary challenge (Figure 4).

The authors then conducted a chemical screen in 3T3-L1 differentiated adipocytes for compounds that modulate lipolysis (glycerol release being the high-throughput assay read-out). They find multiple nitric oxide modulators inhibit lipolysis, and demonstrate that treatment of fasted Tg-3.5ubb:plin2-tdTomato juveniles with the hit compounds also modulates lipid droplet area. These same compounds also modulate lipid droplet number and size in a zebrafish melanoma cell line.

The three reviewers agree that the manuscript represents a potentially major advance. For labs using zebrafish, this tool will provide new abilities to monitor lipid droplets and their potential roles in metabolic challenges and diseases like cancer. This reporter could also allow for novel understanding of heterogeneity of adipocytes and lipid droplets by allowing study of how single cells and even single lipid droplets dynamically change in an organism. The essential revisions were formulated after discussion in the hopes of strengthening the manuscript and yielding a tool of wide interest to the field.

Title:

See the comments below regarding the need to include the chemical screen in the title or abstract.

Essential revisions:

1. A fuller ontogeny of lipid droplets within adipocytes should be feasible with the authors new reporter: starting from feeding (5 to 7 dfp) , a time series of representative larval whole-mount photomicrographs should be presented to demonstrate when, where, and to what extent the reporter begins to report. (Figure 1). Others have used a similar lipid droplet coat protein labeling approach in cell culture models and invertebrates; these citations are more pertinent than several review articles (https://pubmed.ncbi.nlm.nih.gov/12591929/) and (https://pubmed.ncbi.nlm.nih.gov/24894357/). An alternative, probe-free, whole animal imaging platform has been described in zebrafish, and merits discussion: https://pubmed.ncbi.nlm.nih.gov/33268772/.

2. A direct comparison of the tdTomato signal with a fluorescent lipid dye such as BODIPY 493/503 in a time series would allow the reader to grasp what the advantages and disadvantages of both approaches are. A critical control for this experiment is to use analyze the fluorescent lipid signal of casper siblings not carrying the transgene to assess whether the transgene itself alters lipid accumulation. The presence of a Perilipin A transgene in mouse adipose alters whole body physiology, making the need to evaluate the zebrafish Plin2-tdTomato transgenic for possible neomorphic phenotypes critical: https://pubmed.ncbi.nlm.nih.gov/21103377/. Along similar lines, the text from lines 17 to 58 and line 75 should be revised to more fairly describe the body of work using several lipid dyes in this model system.

3. A prolonged fast is feasible, and has been done with fluorescent lipid molecule vital stains (again, see point 1 regarding the robust read-out available with these probes and the need for balanced discussion of their limitations). Conducting such a fast would greatly enhance the results of Figure 2. A time course of up to 7 days would be very informative, revealing how the tdTomato signal decreases as lipid mass decreases. This would also reveal how a transgene under a ubiquitous promoter responds to loss of a cellular structure. This point mirrors the first: casper animals lacking the transgene should be examined with concurrent BODIPY staining, since this would reveal if the transgene's presence alters the kinetics of lipolysis. If single animals could be tracked, that would also enhance the power of the tool to reveal individual organism lipid levels. Of course, repeated microscopic mounting might not be feasible; but sufficient animals will survive the process to make initial and final data points informative (Figure 2).

4. The impact of the short-term (7 days) feeding of the "HFD" on mass and condition factor should be presented. The expansion of visceral lipid area seems rather modest in 7 days; doubling this duration and plotting time series for the changes in SL, mass, and adipose fluorescent area would reveal the dynamic range of this tool in revealing increases in adiposity. A second diet of age-appropriate live feed would also make the results more useful to other workers: the defined diet the authors present might not be widely used by others.

5. The scale and scope of the chemical screen should be described in the title, abstract, or both. The potent ATGL inhibitor Atglistatin has a much larger effect in preventing loss of fluorescent area in a 24-hour drug treatment in vivo than the nitric oxide donor JS-K or the thioredoxin reductase inhibitor does. This might reflect dose more than potency of mechanism: 40 μm vs. 1 uM. A dose range for the nitric oxide modulators is needed to make valid comparisons and to address the power of the transgenic system to serve as a valid readout in future chemical screens (i.e., using these animals and not 3T3L1 cells will reveal both autonomous and non-autonomous mechanisms of adipose lipid accumulation). Others have made a similar observation that merits citation: https://pubmed.ncbi.nlm.nih.gov/26317347/.

6. Paralleling points 1 to 3, the melanoma cell experiments should be done with the BODIPY tracer as a comparator to reveal what is gained with the transgenic reporter. A video of melanoma cells does not add much to this manuscript, since it is the power of the in vivo reporter that is being showcased and could be demonstrated better with addressing the above points.

7. Data Availability

The availability of the Tg(-3.5ubb:plin2-tdTomato) casper line is not disclosed, while a competing interest of the senior author's use of the casper line is. Given the NIH funding listed, this model organism should be deposited in the Zebrafish International Resource Center, and a statement regarding this should be included in the manuscript.

---

## [Author Response]

Essential revisions:1. A fuller ontogeny of lipid droplets within adipocytes should be feasible with the authors new reporter: starting from feeding (5 to 7 dfp) , a time series of representative larval whole mount photomicrographs should be presented to demonstrate when, where, and to what extent the reporter begins to report. (Figure 1). Others have used a similar lipid droplet coat protein labeling approach in cell culture models and invertebrates; these citations are more pertinent than several review articles (https://pubmed.ncbi.nlm.nih.gov/12591929/) and (https://pubmed.ncbi.nlm.nih.gov/24894357/). An alternative, probe-free, whole animal imaging platform has been described in zebrafish, and merits discussion: https://pubmed.ncbi.nlm.nih.gov/33268772/.

We have performed a time-series analysis of expression in both WT siblings as well as the *Tg(-3.5ubb:plin2-tdTomato)* line. This was done at 5, 14, 21, 42dpf and adulthood. In addition, and in response to point #2 below, we simultaneously stained with BODIPY to be able to compare our transgenic with a more established lipid dye. Using previously developed nomenclature for various adipocyte depots in zebrafish (Minchin and Rawls, 2017a), we created heatmaps of expression for both BODIPY and tdTomato (Supplementary Figure 2A) and show representative images (Supplementary Figure 2B). These results demonstrate that the transgenic reporter is first visible at 14dpf, which matches what is seen by BODIPY at this time point. Expression in the transgenic line increases with time, as expected and includes additional depots by adulthood, again matching closely what is seen in the BODIPY stained animals.

We agree with the comment that citation of primary literature, rather than review papers, is appropriate. We have corrected this in the text.

2. A direct comparison of the tdTomato signal with a fluorescent lipid dye such as BODIPY 493/503 in a time series would allow the reader to grasp what the advantages and disadvantages of both approaches are. A critical control for this experiment is to use analyze the fluorescent lipid signal of casper siblings not carrying the transgene to assess whether the transgene itself alters lipid accumulation. The presence of a Perilipin A transgene in mouse adipose alters whole body physiology, making the need to evaluate the zebrafish Plin2-tdTomato transgenic for possible neomorphic phenotypes critical: https://pubmed.ncbi.nlm.nih.gov/21103377/. Along similar lines, the text from lines 17 to 58 and line 75 should be revised to more fairly describe the body of work using several lipid dyes in this model system.

We agree that the use of sibling controls, along with BODIPY, was an essential experiment to rule out a neomorphic effect of the transgene. As noted in point #1 above, we have performed these experiments and see strong concordance between BODIPY staining and tdTomato fluorescence, with no obvious effect of the transgene.

Specifically, in Figure 2C-F, we see no significant difference in BODIPY staining between the WT siblings and the *Tg(-3.5ubb:plin2-tdTomato)* line (average BODIPY area: WT = 0.37 ± 0.04 mm^2^ and *Tg(-3.5ubb:plin2-tdTomato)* = 0.34 ± 0.05 mm^2^ | Mann Whitney test p=0.42). Furthermore, in quantifying the effect across time (see the heatmap in Supplementary Figure 2), we see no difference in the timing of BODIPY staining in WT siblings versus transgenics. We also saw no effect of the transgene on lipolysis during prolonged fasting (see point #3, below). Finally, and as explained in more detail in point #6 below, we also found that the transgene did not affect lipid droplet content in the ZMEL melanoma cell line experiments as well. Thus, while we cannot rule out an effect on the transgene is every single physiologic situation, we do not see an effect during normal development or in the cancer context.

We have modified the text to take these new results into account and put them into better context with the existing literature regarding lipid dyes.

3. A prolonged fast is feasible, and has been done with fluorescent lipid molecule vital stains (again, see point 1 regarding the robust read-out available with these probes and the need for balanced discussion of their limitations). Conducting such a fast would greatly enhance the results of Figure 2. A time course of up to 7 days would be very informative, revealing how the tdTomato signal decreases as lipid mass decreases. This would also reveal how a transgene under a ubiquitous promoter responds to loss of a cellular structure. This point mirrors the first: casper animals lacking the transgene should be examined with concurrent BODIPY staining, since this would reveal if the transgene's presence alters the kinetics of lipolysis. If single animals could be tracked, that would also enhance the power of the tool to reveal individual organism lipid levels. Of course, repeated microscopic mounting might not be feasible; but sufficient animals will survive the process to make initial and final data points informative (Figure 2).

We have now performed this experiment as suggested. The new data is now shown in Figure 3 of the revised manuscript. We first performed the control experiment in which we used WT and *Tg(-3.5ubb:plin2-tdTomato)* siblings at 21dpf, and fasted them for 7 days and stained them with BODIPY (to further address point #2 above). Individual animals were tracked within each group, and the data then pooled for each group. We observed a decrease in BODIPY staining to the same extent in the WT or Tg control (Figure 3A-E). We then further quantified the tdTomato signal at days 0, 2, 5, and 7 during a prolonged 7d fast and saw the expected decrease in adipocyte area (Figure 3F-J). These data again suggest that the presence of the transgene does not affect overall lipolysis after fasting, and that the transgene can be used to track lipid droplets over a prolonged period of time.

4. The impact of the short-term (7 days) feeding of the "HFD" on mass and condition factor should be presented. The expansion of visceral lipid area seems rather modest in 7 days; doubling this duration and plotting time series for the changes in SL, mass, and adipose fluorescent area would reveal the dynamic range of this tool in revealing increases in adiposity. A second diet of age-appropriate live feed would also make the results more useful to other workers: the defined diet the authors present might not be widely used by others.

We have performed this experiment in which we fed either a control/standard diet or high-fat diets and measured area and standard length from 0 to 14 days. This revealed that the majority of the increase in adiposity from high-fat diet occurs by day 7, with relatively little further increase by day 14. Surprisingly, we found little acceleration of standard length in this assay, suggesting the effect of HFD is very specific to adipose tissue. These data are shown in Figure 4A-E. We attempted to measure the mass of the fish in addition to SL and fluorescent area but the small SLs and variable adherence of water to the fish produced inconsistent results.

While we agree that live-feed diets remain common in the field, our facility has converted to a powdered feed diet. In addition, there is no simple way to make a high fat diet addition to a live feed diet. We have repeatedly tried adding egg yolk to the diet (to mimic a high fat diet) and have found that the fish consistently avoid eating it. We did not feel it would be a fair comparison to compare a standard live feed diet to a powdered high fat diet. Thus for these experiments, we used either a standard control diet or high-fat feed diet from Sparos, which mirrors what is increasingly being used in many facilities, including ours.

5. The scale and scope of the chemical screen should be described in the title, abstract, or both. The potent ATGL inhibitor Atglistatin has a much larger effect in preventing loss of fluorescent area in a 24-hour drug treatment in vivo than the nitric oxide donor JS-K or the thioredoxin reductase inhibitor does. This might reflect dose more than potency of mechanism: 40 μm vs. 1 uM. A dose range for the nitric oxide modulators is needed to make valid comparisons and to address the power of the transgenic system to serve as a valid readout in future chemical screens (i.e., using these animals and not 3T3L1 cells will reveal both autonomous and non-autonomous mechanisms of adipose lipid accumulation). Others have made a similar observation that merits citation: https://pubmed.ncbi.nlm.nih.gov/26317347/.

We agree that a more robust dose response curve was needed for the in vivo experiments. We first performed dose-limiting toxicity studies in the fish to find the maximally tolerated dose (MTD) that did not cause death or illness. This was 40µM for Atglistatin, 1µM for Auranofin, and 1µM for JS-K (Supplementary Figure 5A-C). Based on this, we tested these drugs at their MTD in vivo and measured lipid droplet area, and confirmed that only Atglistatin and JS-K had an effect at these doses (Figure 5C-E). Because Atglistatin was very well tolerated at multiple doses, we also measured lipid droplet area after exposure to 1µM, 10µM and 40µM and again found only an effect at the MTD of 40µM (Supplementary Figure 5D-F). These data suggest that for in vivo studies, it is likely that the efficacy of a given drug is likely to be greatest at or near the MTD, which is not surprising given the potential for lipolysis to affect systemic physiology.

We have modified the abstract to better reflect the chemical screen aspect of the paper, and will add in the additional relevant citation.

6. Paralleling points 1 to 3, the melanoma cell experiments should be done with the BODIPY tracer as a comparator to reveal what is gained with the transgenic reporter. A video of melanoma cells does not add much to this manuscript, since it is the power of the in vivo reporter that is being showcased and could be demonstrated better with addressing the above points.

We have performed this control experiment. Analogous to what we did in the sibling fish (points #1 and #2 above), we exposed either control ZMEL-GFP or ZMEL-LD cells to increasing doses of oleic acid, and stained them with the far-red dye Lipidtox (which fluoresces in a range distinct from either GFP or tdTomato). We then quantified median fluorescence intensity (MFI) for far-red fluorescence (from the Lipidtox). Consistent with our in vivo results, we saw no effect of the transgene on accumulation of Lipidtox staining after oleic acid treatment.

To further explore this, we then used FACS to quantify the percentage of tdTomato versus Lipidtox positive cells after oleic acid. Instead of measuring MFI in this assay, cells were gated with a negative control population and anything above that control gate is considered positive, as is common in FACS. This was highly informative. While the Lipid staining clearly is responsive to oleic acid, it did not show a very high dynamic range, in that the various concentrations read out in fairly similar ways. In contrast, the same assay using the tdTomato transgene showed very high dynamic range, essentially mirroring the concentration of oleic acid added to the media.

Taken together, these two experiments demonstrate that the transgene does not exert a neomorphic effect on the melanoma cells, and further demonstrates the potential usefulness of the transgene compared to lipid dyes. A consideration of this data has been added to the manuscript.

7. Data AvailabilityThe availability of the Tg(-3.5ubb:plin2-tdTomato) casper line is not disclosed, while a competing interest of the senior author's use of the casper line is. Given the NIH funding listed, this model organism should be deposited in the Zebrafish International Resource Center, and a statement regarding this should be included in the manuscript.

These fish will be deposited in ZIRC as is standard in the field. However, in some cases ZIRC will not accept all transgenics, depending on their capacity. Should that be the case, we would readily make the animals available to any lab who requests it. In addition, we will make all transgenes available via Addgene.